

# Surface-Based Ku- and Ka-band Polarimetric Radar for Sea Ice Studies

Julienne Stroeve[1,2,3], Vishnu Nandan[1], Rosemary Willatt[2], Rasmus Tonboe[4], Stefan Hendricks[5], Robert Ricker[5], James Mead[6], Marcus Huntemann[5,7], Polona Itkin[8], Martin Schneebeli[9], Daniela Krampe[5], Gunnar Spreen[7], Jeremy Wilkinson[10], Ilkka Matero[5], Mario Hoppmann[5], Robbie Mallett[2] and Michel Tsamados[2]

[1]University of Manitoba, Centre for Earth Observation Science, 535 Wallace Building, Winnipeg, MB, R3T 2N2, Canada

[2]University College London, Earth Science Department, Gower Street, WC1E 6BT, UK

[3]National Snow and Ice Data Center, University of Colorado, 1540 30th Street, Boulder, CO 80302, USA

[4]Danish Meteorological Institute, Lyngbyvej 100, 2100 Copenhagen, Denmark

[5]Alfred Wegener Institute, Am Handelshafen 12, 27570 Bremerhaven, Germany

[6]ProSensing, 107 Sunderland Road, Amherst, MA, 01002-1357, USA

[7]Institute of Environmental Physics, University of Bremen, Otto-Hahn-Allee 1, D-28359 Bremen, Germany

[8]UiT The Arctic University of Norway, Department of Physics and Technology, Tromsø, 9019, Norway

[9]WSL Institute for Snow and Avalanche Research SLF, Fluelastrasse 11, CH-7260 Davos Dorf, Switzerland

[10]British Antarctic Survey, High Cross, Madingley Road, Cambridge, CB30ET, UK

*Correspondence to*: Julienne Stroeve (julienne.stroeve@umanitoba.ca)

**Abstract.** To improve our understanding of how snow properties influence sea ice thickness retrievals from presently operational and upcoming satellite radar altimeter missions, as well as investigating the potential for combining dual frequencies to simultaneously map snow depth and sea ice thickness, a new, surface-based, fully-polarimetric Ku- and Ka-band radar (KuKa radar) was built and deployed during the 2019-2020 year-long MOSAiC International Arctic drift expedition. This instrument, built to operate both as an altimeter (stare mode) and a scatterometer (scanning mode), provided the first *in situ* Ku- and Ka-band dual frequency radar observations from autumn freeze-up through mid-winter, and covering newly formed ice in leads, first-year and second-year ice floes. Data gathered in the altimeter mode, will be used to investigate the potential for estimating snow depth as the difference between dominant radar scattering horizons in the Ka- and Ku-band data. In the scatterometer mode, the Ku- and Ka-band radars operated under a wide range of azimuth and incidence angle ranges, continuously assessing changes in the polarimetric radar backscatter and derived polarimetric parameters, as snow properties varied under varying atmospheric conditions. These observations allow for characterizing radar backscatter responses to changes in atmospheric and surface geophysical conditions. In this paper, we describe the KuKa radar and illustrate examples of these data and demonstrate their potential for these investigations.



## 1 Introduction

Sea ice is an important indicator of climate change, playing a fundamental role in the Arctic energy and freshwater balance. Furthermore, because of complex physical and biogeochemical interactions and feedbacks, sea ice is also a key component of the marine ecosystem. Over the last several decades of continuous observations from multi-frequency satellite passive

microwave imagers, there has been a nearly 50% decline in Arctic sea ice extent at the time of the annual summer minimum (Stroeve and Notz, 2018; Stroeve et al., 2012; Parkinson and Cavalieri, 2002; Cavalieri et al., 1999). This loss of sea ice area has been accompanied by a transition from an Arctic Ocean dominated by older and thicker multi-year ice (MYI) to one dominated by younger and thinner first-year ice (FYI) (Maslanik et al., 2007, 2011). While younger ice tends to be thinner and more dynamic, much less is known about how thickness and volume are changing. Accurate ice thickness monitoring is

essential for heat and momentum budgets, ocean properties and timing of sea ice algae and phytoplankton blooms (Bluhm et al., 2017; Mundy et al., 2014).

Early techniques to map sea ice thickness relied primarily on *in situ* drilling, ice mass balance buoys, upward looking sonar on submarines and moorings, providing limited spatial and temporal coverage, and have been logistically difficult. More recently, electromagnetic systems, including radar and laser altimeters flown on aircraft and satellites, have expanded

these measurements to cover the pan-Arctic region. However, sea ice thickness is not directly measured by laser or radar altimeters. Instead these types of sensors measure the ice or snow freeboard, which when combined with assumptions on the amount of snow on the ice, radar penetration of the surface, and the snow, ice and water densities, can be converted into total sea ice thickness assuming hydrostatic equilibrium (Laxon et al., 2003; Laxon et al., 2013; Wingham et al., 2006; Kurtz et al., 2009).

Current satellite-based radar altimeters, such as the European Space Agency (ESA)'s Ku-band CryoSat-2 (CS2) since April 2010, and Ka-band SARAL/AltiKa, launched in February 2013 as part of a joint mission by the Centre National d'Etudes Spatiales (CNES) and the Indian Space Research Organization (ISRO), provide the possibility to map pan-Arctic (up to 81.5° N for AltiKa) sea ice thickness (Tilling et al., 2018; Hendricks et al., 2016; Kurtz and Harbeck, 2017; Armitage and Ridout, 2015). It may also be possible to combine Ku- and Ka-bands to simultaneously retrieve both ice thickness and

snow depth during winter (Lawrence et al., 2018; Guerreiro et al., 2016). Other studies have additionally suggested the feasibility of combining CS2 with snow freeboard observations from laser altimetry (e.g. ICESat-2) to map pan-Arctic snow depth and ice thickness, during the cold season (Kwok and Markus, 2018; Kwok et al., 2020).

However, several key uncertainties limit the accuracy of the radar-based freeboard retrieval, which then propagate into the freeboard-to-thickness conversion. One important uncertainty pertains to inconsistent knowledge on how far the radar

signal penetrates into the overlying snow cover (Nandan et al., 2020; Willatt et al., 2011; Drinkwater, 1995a). The general assumption is that the radar return primarily originates from the snow/sea ice interface at Ku-band (CS2), and from the air/snow interface at Ka-band (AltiKa). While this may hold true for cold, dry snow in a laboratory (Beaven et al., 1995),




scientific evidence from observations and modelling suggests this assumption may be invalid even for a cold, homogeneous snowpack (Nandan et al., 2020; Willatt et al., 2011; Tonboe et al., 2010). Modelling experiments also reveal that for every

mm of SWE, the effective scattering surface is raised by 2 mm relative to the freeboard (Tonboe, 2017). A further complication is that radar backscattering is sensitive to the presence of liquid water within the snowpack. This means that determining the sea ice freeboard using radar altimeters during the transition phase into Arctic summer is not possible (Beaven et al., 1995; Landy et al., 2019). The transition from a MYI- to FYI-dominated Arctic has additionally resulted in a more saline snowpack, which in turn impacts the snow brine volume, thereby affecting snow dielectric permittivity. This

vertically shifts the location of the Ku-band radar scattering horizon by several centimetres above the snow/sea ice interface (Nandan et al., 2020; Nandan et al., 2017b; Tonboe et al., 2006). As a result, field campaigns have revealed that the dominant radar scattering actually occurs within the snowpack or at the snow surface rather than at the snow/ice interface (Willatt et al., 2011; Giles et al., 2007). Another complication is that surface roughness and sub-footprint preferential sampling may also impact the location of the main radar scattering horizon (Tonboe et al., 2010; Landy et al., 2019). All

these processes combined result in significant uncertainty as to accurately detecting the location of the dominant Ku-band scattering horizon, and in turn influence the accuracy of sea ice thickness retrievals from satellites. This would also create biases in snow depth retrievals obtained from combining dual frequency radar observations or from combining radar and laser altimeter observations, as recently done in Kwok et al. (2020).

   Other sources of error in radar altimeter sea ice thickness retrievals include assumptions on ice, snow and water

densities used in the conversion of freeboard to ice thickness, inhomogeneity of snow and ice within the radar footprint, and snow depth. Lack of snow depth and snow water equivalent (SWE) knowledge provides the largest uncertainty (Giles et al., 2007). Yet, snow depth is not routinely measured by satellites despite efforts to use multi-frequency passive microwave brightness temperatures to map snow depth over FYI (Markus et al., 2011), and also over MYI (Rostosky et al., 2018). Instead, climatological values are often used, based on data collected several decades ago on MYI (Warren et al., 1999;

Shalina and Sandven, 2018). These snow depths are arguably no longer valid for the first-year ice regime which now dominates the Arctic Ocean (70% FYI today vs. 30% in 1980s). To compensate, radar altimeter processing groups have halved the snow climatology over FYI (Tilling et al., 2018; Hendricks et al., 2016; Kurtz and Farrell, 2011), yet climatology does not reflect actual snow conditions on either FYI or MYI for any particular year and also not the spatial variability at the resolution of a radar altimeter. The change in ice type, combined with large delays in autumn freeze-up and earlier melt

onset (Stroeve and Notz, 2018) have resulted in a much thinner snowpack compared to that in the 1980s (Stroeve et al., 2020; Webster et al., 2014). The use of an unrepresentative snow climatology can result in substantial biases in total sea ice thickness, if the snow depth departs strongly from this climatology. Moreover, snow depth is also needed for the radar propagation delay in the freeboard retrieval and for estimating snow mass in the freeboard to thickness conversion. If snow depth is unknown and climatology is used instead, error contributions are stacked and amplified when freeboard is converted

to ice thickness. Therefore, the potential to combine Ku- and Ka-bands to map both snow depth, radar penetration and ice



thickness at radar footprint resolution is an attractive alternative and forms one of the deltas of a possible follow-on mission to CS2, such as the ESA's Copernicus candidate mission CRISTAL (Kern et al., 2020).

Besides altimeters, active radar remote sensing has proven its capability to effectively characterize changes in snow/sea ice geophysical and thermodynamic property conditions, at multiple microwave frequencies (Barber and Nghiem, 1999;
Drinkwater, 1989; Gill et al., 2015; Komarov et al., 2015; Nandan et al., 2016; Nandan et al., 2017a). Snow and its associated geophysical and thermodynamic properties play a central role in the radar signal propagation and scattering within the snow-covered sea ice media (Barber and Nghiem 1999; Nandan et al., 2017a; Barber et al., 1998; Yackel and Barber, 2007; Nandan et al., 2020). This in turn impacts the accuracy of satellite-derived estimates of critical sea ice state variables, including sea ice thickness, snow depth, SWE, and timings of melt-, freeze- and pond-onset.

At Ku- and Ka-bands, currently operational and upcoming Synthetic Aperture Radar (SAR) missions operate over a wide range of polarizations, spatial and temporal resolutions and coverage area. Due to the presence of possible spatial heterogeneity of snow and sea ice types present within a satellite resolution grid cell, the sensors add significant uncertainty to direct retrievals of snow/sea ice state variables. In addition, radar signals acquired from these sensors may be temporally de-correlated, owing to dynamic temporal variability of snow and sea ice geophysical and thermodynamic properties. To
avoid this uncertainty, high spatial and temporal resolution *in situ* measurements of radar backscatter from snow-covered sea ice are necessary, quasi-coincident to unambiguous *in situ* measurements of snow/sea ice geophysical and thermodynamic properties (Nandan et al., 2016; Geldsetzer et al., 2007). Although, a wide range of research has utilized dual- and multi-frequency microwave approaches to characterize the thermodynamic and geophysical state of snow-covered sea ice, using surface-based and airborne multi-frequency, multi-polarization measurements (Nandan et al., 2016; Nandan et al., 2017a;
Beaven et al., 1995; Onstott et al., 1979; Livingstone et al., 1987; Lytle et al., 1993), no studies have been conducted using coincident dual-frequency Ku- and Ka-band radar signatures of snow-covered sea ice to investigate the potential of effectively characterize changes in snow/sea ice geophysical and thermodynamic properties, with variations in atmospheric forcing.

From a radar altimetry standpoint, there are differences in scattering mechanisms from surface- and satellite-based
systems. From a satellite-based system, the radar backscatter is dominated by surface scattering, while for a surface-based radar system, the backscatter coefficient is much lower, because the surface-based system is not affected by the high coherent scattering from large facets (large relative to the wavelength) within the Fresnel reflection zone (Fetterer et al., 1992). In addition, observations from ground-based radar systems can target homogenous surfaces and thus directly interpret coherent backscatter contribution of the various surface types which are often mixed in satellite observations, which requires
backscatter decomposition. Therefore, it is important to study the Ku- and Ka-band radar propagation and behavior in snow-covered sea ice, using surface-based systems and how they can be used for understanding scattering from satellite systems.





To improve our understanding of snowpack variability on the dominant scattering horizon relevant to satellite radar altimetry studies, as well as backscatter variability for scatterometer systems, a Ku- and Ka-band dual-frequency, fully-polarimetric radar (KuKa radar) was built and deployed during the year-long Multidisciplinary drifting Observatory for the

Study of Arctic Climate (MOSAiC) International Arctic drift expedition (https://mosaic-expedition.org/expedition/). The KuKa radar provides a unique opportunity to obtain a benchmark dataset, involving coincident field, airborne and satellite data, from which we can better characterize how the physical properties of the snow pack (above different ice types) influence the Ka- and Ku-band backscatter and penetration. Importantly, for the first time we are able to evaluate the seasonal evolution of the snowpack over FYI and MYI. MOSAiC additionally provides the opportunity for year-round

observations of snow depth and its associated geophysical and thermodynamic properties, that will allow for rigorous assessment of the validity of climatological assumptions typically employed in thickness retrievals from radar altimetry as well as providing data for validation of snow depth products. These activities are essential, if we are to improve sea ice thickness retrievals and uncertainty estimation from radar altimetry over the many ice and snow conditions found in the Arctic and the Antarctic.

This paper describes the KuKa radar and its deployment during the MOSAiC drift expedition, including some initial demonstration of fully-polarimetric data (altimeter and scatterometer modes) collected over different ice types from mid-October 2019 through the end of January 2020.

## 2    The Ku- and Ka-band dual frequency system

Given the importance of snow depth on sea ice thickness retrievals from satellite radar altimetry, several efforts are

underway to improve upon the use of a snow climatology. One approach is to combine freeboards from two satellite radar altimeters of different frequencies, such as AltiKa and CS2, to estimate snow depth (Lawrence et al., 2018; Guerreiro et al., 2016). Early studies comparing freeboards from these two satellites showed AltiKa retrieved different elevations over sea ice than did CS2 (Armitage and Ridout, 2015), paving the way forward for combining these satellites to map snow depth. However, freeboard differences showed significant spatial variability and suggested Ka-band signals are sensitive to

surface/volume scattering contributions from the uppermost snow layers, and sensitivity of Ku-band signals to snow layers that are saline and complexly-layered (via rain-on-snow and melt-refreeze events). These complexities in snow properties largely impact the Ka- and Ku-band radar penetration depth. Penetration depths at Ka- and Ku-band evaluated against NASA's Operation Ice Bridge (OIB) freeboards found mean penetration factors (defined as the dominant scattering horizon in relation to the snow and ice surfaces) of 0.45 for AltiKa and 0.96 for CS2 (Armitage and Ridout, 2015). A key limitation

of this approach however is that, it is based on OIB data that cover a limited region of the Arctic Ocean and are only available during springtime. OIB snow depths also have much smaller footprints than the large footprints of CS2/AltiKa. Further, this approach assumes that the OIB-derived snow depths are correct.



Biases from sampling differences, potential temporal decorrelation between different satellites and processing techniques also play a role. With regards to combining AltiKa and CS2, the larger AltiKa pulse-limited footprint compared

to the CS2 beam-sharpening leads to different sensitivity to surface roughness for the differences due to the different footprint sizes illuminating a different instantaneous surface. This approach is further complicated by the fact that the satellite radar pulses have travelled through an unknown amount of snow, slowing the speed of the radar pulse, leading to radar freeboard retrievals that differ from actual sea ice freeboards. Other sources of biases in the radar processing chain include (i) uncertainty of the return pulse retracking, (ii) off-nadir reflections from leads or 'snagging', (iii) footprint

broadening for rougher topography and (iv) surface type mixing in the satellite footprints.

## 3   Methods

### 3.1 The KuKa Radar

Sea ice thickness is not directly measured by laser or radar altimeters. Instead, sensors such as CS2 retrack the return waveform based on scattering assumptions and from that the ice freeboard ($f_i$) can be derived. This can be converted to ice

thickness ($h_{ice}$) assuming hydrostatic equilibrium together with information on snow depth ($h_{snow}$), snow density ($\rho_{snow}$), ice density ($\rho_{ice}$) and water density ($\rho_{water}$) following equation 1:

$$h_{ice} = \frac{\rho_{snow} h_{snow} + \rho_{water} f_i}{\rho_{water} - \rho_{ice}}$$   (Eq. 1)

Snow and ice density are not spatially homogeneous: sea ice density is related to the age of the ice (FYI vs. MYI), while snow density can cover a large spectrum of values depending on weather conditions and heat fluxes.  How far the radar

signal penetrates into the snowpack determines $f_i$, which depends on the dielectric permittivity ($\varepsilon$) of the snowpack, or the ability of the snowpack to transmit the electric field (Ulaby et al., 1986)  and the scattering in the snowpack from the snow microstructure and scattering at the air/snow, snow/sea ice and internal snow layers. The permittivity can be written as $\varepsilon = \varepsilon' + i\varepsilon''$, where $\varepsilon'$ is the real part of the permittivity and $\varepsilon''$ is the imaginary part, and depends on $\rho_{snow}$ and frequency of the radiation penetrating through the snowpack: the higher $\varepsilon''$, the more the field strength is reduced (absorption). Dry snow

is a mixture of ice and air, and therefore its complex permittivity $\varepsilon$ depends on the dielectric properties of ice, snow microstructure and snow density (Ulaby et al., 1986). In general,  dry snow permittivity scales linearly with $\rho_{snow}$, such that increasing $\rho_{snow}$ increases $\varepsilon'$ (Ulaby et al., 1986). A further complication is that radar backscattering is sensitive to the presence of liquid water and brine within the snowpack (Tonboe et al., 2006; Hallikainen, 1977), such that $\varepsilon'$ for water inclusions is 40 times larger than for dry snow, decreasing the depth to which the radar will penetrate. In other words, small

amounts of liquid water lead to lower penetration depth (Winebrenner et al., 1998). Negative freeboards can additionally lead to snow flooding creating a slush layer and wicking up of moisture. These can all lead to the presence of moisture in the snow pack even in winter months when the air temperature would indicate that the snow was cold and dry, and hence, the dominant scattering surface in the Ku-band would assumed to be the snow/ice interface (Beaven et al., 1995). The processes listed here determine the shape of the radar altimeter waveform and the subsequent impact on the freeboard depends on the



retracker algorithm applied on the altimeter waveform, to determine the location of the main radar backscatter horizon (e.g. Ricker et al., 2014).

When developing an *in situ* radar system to study radar penetration into the snowpack, it is important to consider how the snow dielectric permittivity, surface and volume scattering contributions to the total backscatter changes temporally (both diurnally and seasonally), as new snow accumulates and is modified by wind redistribution, temperature gradients, and
salinity evolution over newly formed sea ice. Surface scattering dominates from dielectric interfaces such as air/snow, internal snow layers and from snow/sea ice interface, while volume scattering dominates from the snow microstructure or from inclusions within the ice (Ulaby et al. 1986). For snow and ice surfaces, surface scattering dominates (i.e. from the snow surface, from the ice surface, and from internal snow layering). Because snow is a dense media, scattering from individual snow grains is affected by its neighbors and the volume scattering is not simply the non-coherent sum of all
scatterers, but must include multiple scattering effects. With surface-based radar systems, it is important to understand what kind of scattering mechanisms are to be expected from the snow/sea ice media.

To resolve the scattering properties of snow from the surface and subsurface layers, the new KuKa radar designed by ProSensing Inc. was configured to operate both as an altimeter and as a scatterometer. Built for Arctic conditions, the KuKa radar transmits at Ku- (12-18 GHz) and at Ka-bands (30-40 GHz) using a very low power transmitter, making it suitable for
short ranges (typically less than 30 m). Both Ku- and Ka-band radio frequency (RF) units are dual-polarization, solid-state FMCW (frequency modulated continuous wave) radars using linear FM modulation. Each system employs a linear FM synthesizer with variable bandwidth for two modes, fine and coarse range resolution. The system is configured to always operate in fine mode, with a bandwidth of 6 and 10 GHz at Ku- and Ka-bands, respectively, but any segment of the 12-18 GHz or 30-40 GHz bandwidth can be processed to achieve any desired range resolution above 2.5 cm (Ku-band) or 1.5 cm
(Ka-band). Coarse range resolution processing is centered on satellite frequencies of CS2 and AltiKa (e.g. 13.575 GHz and 35.7 GHz, respectively), with an operating bandwidth of 500 MHz, yielding 30 cm range resolution. Polarization isolation of the antennas is greater than 30 dB. An internal calibration loop, consisting of an attenuator and 4.2 m long delay line (electrical delay = 20 ns) is used to monitor system stability. This calibration loop data is used in the data processing software to compensate for any power drift as a result of temperature changes. During the polar winter, air temperatures
regularly drop to -30°C to -40°C, while cyclones entering the central Arctic can result in air temperatures approaching 0°C during mid-winter (Graham et al., 2017). The RF units are insulated and heated to stabilize the interior temperature under such cold conditions. Given that this instrument was designed for polar conditions, it is not intended to operate at temperatures above 15°C. Operating parameters for each RF unit are summarized in **Table 1**.

The antennas of each radar are dual-polarized scalar horns with a beamwidth of approximately 16.5° at Ku-band and
11.9° at Ka-band. Thus, they are not scanning exactly at the same surface because of slightly different footprints. However, the different footprint sizes of each band are to some extent averaged out by the spatial/temporal averaging (discussed in



section 2.3). Further, they do not take data at the same rate. At Ku-band, a new block of data is gathered every 0.5s, while at Ka-band a new block of data is gathered every 0.33s. Also, the GPS data is independent between the two instruments, so any random drift in the latitude/longitude can have a small effect on the estimated position. Further, data acquisition is not

precisely time-aligned between the two instruments; start times vary by ~ 0.5s. The radar employs a fast linear FM synthesizer and pulse-to-pulse polarization switching, which allows the system to measure the complex scattering matrix of a target in less than 10 ms. This allows the scattering matrix to be measured well within the decorrelation distance (approximately half the antenna diameter) when towing the radar along the transects path at 1-2 m/s.

During the MOSAiC field campaign, the radar was operated in both in a nadir "*stare*" (or altimeter) mode as well as in a

"*scan*" (or scatterometer) mode when attached to a pedestal that scans over a programmed range of azimuth and incidence angles ($\theta$) (See **Figure 1**). In this configuration, the radar and positioner were powered by 240 VAC 50 Hz power to the input of the UPS mounted on the pedestal. For the altimeter mode, the RF units were unmounted from the positioner and attached to a ridge frame attached to a transect sled. Two 12 VDC batteries were used to power the RF units during the stare mode.

In stare/transect mode, the radar measures the backscatter at nadir ($\theta = 0°$) as a function of time. In stare mode, a new file is generated and stored every 5 minutes. The radar data was processed in segments based on the lateral travel distance of the sledge where the instrument was placed. Given the radar antenna diameters (0.15 m for Ku and 0.09 m for Ka), the lateral distance traveled by the sledge needs to be 0.5 times the antenna diameters, or 0.075 m and 0.045 m for Ku- and Ka-bands, respectively. The minimum velocity was set to 0.4 m/s to avoid drifting GPS location from appearing as true motion.

In the scatterometer mode, both the Ka- and Ku-band scatterometer beam scans at the programmed $\theta$, moving across the azimuth within a prescribed azimuthal angular width. The system then moves up to the next $\theta$ at a set of increment (for e.g. 5° used for our measurements), and scans the next elevation line along the same azimuthal angular width. A new file, each for Ku- and Ka-bands is generated each time the positioner begins a scan. The footprint of the KuKa radar during one complete scan is a function Ku- and Ka-band antenna beamwidth, and the system geometry, with the footprint increasing in

area, as incidence angle increases from nadir- to far-range.  At ~1.5 m (positioner + pedestal + sledge) height, the KuKa footprint is ~ 15 cm at nadir and ~ 90 cm (Ku-band) and ~ 70 cm (Ka-band) at 50°. With 5° increments in $\theta$ steps, there is an ~ 60% (Ka-band) to 70% (Ku-band overlap within the adjacent incidence angle scans. The number of independent range gates at nadir is about 6 (Ku-band) and 10 (Ka-band), and at 50° incidence angle, the range gates are about 36 (Ku-band) and 46 (Ka-band). The number of Ka- and Ku-band independent samples was obtained by dividing the azimuthal angular width

(90°) by half of the antenna beamwidth and multiplying it by the number of range gates falling within the scatterometer footprint. Based on the range gates, at nadir and at 50° incidence angle, the KuKa radar produces 162 (nadir) and 450 (50°), and 972 (nadir) and 2070 (50°) independent samples, for Ku- and Ka-bands, respectively. Detailed description of range gate



and independent samples calculation can be found at King et al. (2013) and Geldsetzer et al. 2007. No near-field correction is needed since the far field distance is 1m. Polarimetric calibration information is provided in the Supplemental Material.

Since snow consists of many small individual scatterers and scattering facets, with each scatterer having a scattering coefficient, the radar pulse volume consists of a large number of independent scattering amplitudes depending on the size of the antenna and the radar footprint, the size, roughness and slope of the scattering facets and the size and shape of snow and ice scatterers, i.e. snow structure and air-bubbles or brine pockets in the ice. Thus, any particular radar sample received by the RF unit consists of a complex sum of voltages received from all individual scatterers facets as well as multiple

interactions among these. Regardless of the distribution of the scattering coefficients, the fact that they are at different ranges from the antenna gives rise to a random-walk sum, which exhibits a bivariate Gaussian distribution in the complex voltage plane. The power associated with the bivariate Gaussian distribution has a Rayleigh distribution, with a large variance. Thus, to reduce the variance, the radar sweeps across several azimuthal angles, or in the case of nadir view, across a specified distance. There is always a tradeoff between getting enough averaging to converge to the correct mean value for all of the

polarimetric values measured by the radar for enhanced range resolution while avoiding too much spatial averaging.  For the nadir view, the minimum distance travelled to ensure statistically independent samples is half of the antenna diameter. An onboard GPS was used to track the radar location, and sample values were only included in the final average if the antenna had moved at least half a diameter from the previously included data samples.

    The system can be operated remotely through the internet using the Wide Area Network connection provided. Raw data

is stored on the embedded computer for each RF unit. A webpage allows the user to monitor system operation, configure the scanning of the radar, set up corner reflector calibration, manually move the positioner as well as manage and download the raw data files.

**3.2 KuKa radar Setup and Deployment**

    The MOSAiC Central Observatory (CO) around the German research vessel *R/V Polarstern* was established on an oval

shaped ice floe approximately 3.5 km by 2.5 km. This floe was heavily deformed, and consisted of predominantly remnant second-year ice (SYI). The ridged (or thick) part of the floe was called the "fortress" where all permanent installations were placed. At the beginning of the floe set up, the bottom of the ice was rotten, with only the top 30 cm solid. Melt pond fraction was greater than 50%. The first deployment of the KuKa radar was on 18 October 2019 at Remote Sensing (RS) site (**Figure 2**), on a section of the ice that was approximately 80 cm thick.  However, the ice pack was quite dynamic and a

large storm on 16 -18 November caused break-up of the CO and all RS instruments were turned off and moved to a temporary safe location. On 26 November, the complete RS site was moved closer to MET city (atmospheric meteorological station), on a refrozen melt pond a site also with about 80 cm thick ice, but overall the snow was slightly deeper. The instrument was redeployed on 29 November until 12 December when several leads formed and all instruments were once

again moved to thicker ice and turned off. The KuKa radar started measuring again on 21 December 2019 until 31 January

2020, after which the radar was taken off the RS site to conduct maintenance.

Characterization of the spatial and temporal evolution of Ku- and Ka-band radar penetration into the snow was achieved with two configurations of the radar: 1) near-hourly (55 min) scanning across 90° azimuth and incidence angles between 0° and 50° at 5° increments, at RS City and 2) repeated weekly transects of 1-8 km in length in nadir-stare mode.

Since the internal structure of the snowpack determines its scattering properties (i.e. permittivity, scatter size), detailed

weekly snow pit observations were obtained as close as possible to the RS site. These observations included snow specific surface area (SSA), the scatter correlation length (Proksch et al., 2015) and density made using a SnowMicroPen (SMP), snow/air and snow/ice interface temperatures with a temperature probe, snow salinity with a salinometer and SWE using a 50 cm metal ETH tube together with a spring scale. In case of hard crusts too hard for the SMP to work, snow density was collected using a density cutter. In addition to these basic snow pit measurements, near-infrared (NIR) photography and

micro-CT scanning were also conducted. The NIR camera allows for determination of snow layers with different SSA at a spatial resolution of about 1 mm (Matzl and Schneebeli, 2006). MicroCT scanning on the other hand provides 3D details on snow microstructure using X-ray microtomography. A thermal infrared (TIR) camera (Infratec VarioCam HDx head 625) was set up spatially observe the surface temperature of the entire remote sensing footprint at regular 10-minute intervals. The setup was supported by a visual surveillance camera taking pictures at 5-minute intervals to resolve event, such as snow

accumulation and formation of snow dunes. During leg 2 of the MOSAiC expedition (i.e. 15 December 2019 through 22 February 2020), ice cores and freeboard observations were occasionally collected near the RS instruments, and bi-weekly snow depths were measured around each instrument. Finally, two digital thermistor strings (DTCs) were installed at the RS site and provided additional information on temperature profiles within the snow and ice (at 2 cm vertical resolution), from which snow depth and sea ice thickness can be inferred.

For the stare/transect mode, nadir-view radar measurements were collected in parallel with snow depth from MagnaProbe (rod of 1.2 m in length, https://patents.google.com/patent/US5864059A/en) equipped with GPS, and a ground-based Broadband Electromagnetic Induction Sensor for total ice thickness (Geophex GEM-2). The CO included both a Northern and a Southern transect loop (**Figure 2**), with the northern loop representing thicker and rougher ice and the southern loop representing younger and thinner ice that has been formed in former melt ponds. Snow pit measurements were

collected along a portion of the Northern transect, at typically six select locations spaced ~100 m apart. At each pit, SMP measurements provided SSA and snow density information (5 measurements at each location), together with snow/air and snow/ice interface temperatures, snow salinity and SWE.

While these data were routinely collected to support interpretation of the radar backscatter, snow on sea ice is spatially variable at a variety of scales as wind redistribution results in the formation of snow dunes and bedforms (Moon et al., 2019;

Filhol and Sturm, 2015). Further, different ice types (i.e. FYI vs MYI) have different temporal evolutions of snow depth. In



recognition of the spatially and temporally varying snowpacks, other detailed snow pits were made over different ice conditions, including ridged ice, newly formed lead ice with snow accumulation, level FYI and MYI, and refrozen melt ponds. The key requirement was to adapt the snow sampling to these situations and sampling after significant snowfall and/or snow redistribution. This was especially important for the transect data which sampled several snow and ice types not

represented by the six snow pits. All these data collected in tandem with the KuKa radar will enable in depth investigations of how snow pack variability influences the radar backscatter.

This paper focuses on showing examples of the data collected during the first 3½ months of operation (18 October 2019 through 31 January 2020 during MOSAiC Legs 1 and 2), at both *scan* (scatterometer) and *stare* (altimeter) modes. In depth analysis of how snow pack properties influences the dual-frequency radar returns will form follow-on papers. Nevertheless,

we show examples for different ice types and under different atmospheric conditions. Air temperatures between October and January fluctuated between -5°C and -35°C as measured on the ship (**Figure 3a**), while the ice surface temperature measurements via the TIR camera and the DTC (**Figure 4**) were usually colder than the ship temperatures. During this time, a total number of 18 transect/stare mode operations of the KuKa were made. **Table 3** summarizes the dates over which the transects were made, as well as other opportune sampling. We should note that during Leg 1, only two short northern loop

transects that covered the remote sensing section were sampled. In addition, one frost flower event was sampled over 10 cm thin ice. During Leg 2, the team made weekly transects each week starting 19 December 2019 until the KuKa radar was taken off from the ice for maintenance. In addition, the team made two transects over FYI along the "runway" built on the portside of the ship, and two lead transects spaced a day apart.

In the results section, we highlight results during a relatively warm and cold time-period to see how air and snow

surface temperature influences the Ku- and Ka-band polarimetric backscatter and derived polarimetric parameters at the RS site; November 10 and 15, where the air (snow) temperatures were -28°C (-28°C) and -12°C (-8°C), respectively (**Figures 3 and 4**). For the transects, we show preliminary results for the northern, southern and lead transects in order to highlight different snow/ice types. **Figure 5** summarizes snow depth distributions for the northern (**Figure 5a**) and southern (**Figure 5b**) transects during January, respectively. Overall, the snow was deeper over SYI that was the dominant ice type for the

northern transect compared to the southern transect which consisted in part also of FYI. Mean snow depths for the northern and southern transects ranged from 24.2 cm to 26.7 cm and 19.6 cm to 22.2 cm, respectively from 2 January to 30 January.

### 3.3 Radar data processing

During data acquisition, the KuKa radar acquires data on a series of six signal states: the four transmit polarization combinations (VV, HH, HV and VH), a calibration loop signal and a noise signal. Each data block consists of these six

signals and are processed separately for each frequency. Data are processed into range profiles of the complex received voltage, through fast fourier transform (FFT). The range profiles for each polarization combination are power-averaged in azimuth for each incidence angle. In *stare* mode, the range profiles, gathered at nadir, are spatially averaged with 20



independent records averaged to reduce variance. For the *scan* mode, this procedure is done across the entire azimuthal angular width, for every incidence angle, $\theta$. From the averaged power profiles, the Ku- and Ka-band radar cross section per

unit area (NRCS) is calculated following (Sarbandi et al., 1990), to obtain co-polarized ($\sigma_{VV}^0$ and $\sigma_{HH}^0$) and cross-polarized ($\sigma_{HV}^0$ and $\sigma_{VH}^0$; with $\sigma_{HV}^0 \sim \sigma_{VH}^0$ assuming reciprocity) backscatter cross sections. The polarimetric parameters: co-polarized ratio ($\gamma_{CO}$), cross-polarized ratio ($\gamma_{CROSS}$), co-polarized correlation coefficient ($\rho_{VVHH}$) and co-polarized phase difference ($\varphi_{VVHH}$) are also derived along with the polarimetric backscatter from the average covariance matrix (derived from the complex scattering matrix), of all azimuthal data blocks, within every incidence angle scan line, given by:


$$\text{Co-pol ratio } \gamma_{CO} = \frac{\sigma_{VV}^0}{\sigma_{HH}^0} \qquad \text{(Eq. 2)}$$

$$\text{Cross-pol ratio } \gamma_{CROSS} = \frac{\sigma_{HV}^0}{\sigma_{HH}^0} \qquad \text{(Eq. 3)}$$

$$\text{Co-polarized correlation coefficient } \rho_{VVHH} = \left| \frac{\langle S_{HH}S_{VV}^* \rangle}{\sqrt{\langle S_{HH}S_{HH}^* \rangle \langle S_{VV}S_{VV}^* \rangle}} \right| \qquad \text{(Eq. 4)}$$

$$\text{Co-polarized phase difference } \varphi_{VVHH} = \tan^{-1}\left[ \frac{Im\langle S_{HH}S_{VV}^* \rangle}{Re\langle S_{HH}S_{VV}^* \rangle} \right] \qquad \text{(Eq. 5)}$$

where $S_{ij}$ are complex scattering matrix elements.

      The linear FM signal for each polarization state has a duration of 2 ms, followed by a 100 ns gap. Thus, the total time required to gather the data used in computing the complex received voltages is 8.3 ms. To assure proper estimation of the co-polarized correlation coefficient and phase difference, it is important that the antenna moves much less than half an antenna diameter during the time period between the VV and HH measurements (2.1 ms). Using an allowable movement of 1/20 of

antenna diameter in 2.1 ms, the maximum speed of the sled during the nadir measurements is limited to approximately 2.1 m/s at Ka-band and 3.5 m/s at Ku-band. The software provided by ProSensing converts the Ku- and Ka-band raw data in both *stare* and *scan* modes, into calibrated polarimetric backscatter and parameters of the target covariance matrix and/or Mueller matrix. The Ku- and Ka-band signal processing, calibration, derivation of polarimetric backscatter and parameters, near-field correction and system error analysis are implemented similar to the C- and X-band scatterometer processing, built

and implemented by ProSensing, and described in detail by Geldsetzer et al. (2007) and King et al. (2013), respectively.

      An experiment was done to investigate the response of the internal calibration loop in comparison to the instrument response when a metal plate was placed on the surface. This serves as a vertical height reference for the radar returns, and demonstrates the response of the system to a flat, highly-scattering surface. **Figure 6** shows the experiment conducted with the metal plate for the Ka-band **(Figure 6(a))** and Ku-band **(Figure 6(b))**. The metal plate and calibration loop data are

consistent and in good agreement with each other (black and red, respectively), which indicates that the shape of the return





including internal reflections are well characterized in the calibration data. The blue data show the scattering from the exposed snow and ice (prior to placing the metal plate). The range of the peak is slightly larger than for the metal plate data, this could be because the metal plate, approximately 15 × 55 cm in size, did not fill the entire footprints of the Ka- and Ku-band antennas. Therefore, its surface appears closer than the snow surface as it dominates the return: the measured peak

range of the metal plate of 1.53 m; when the plate is removed, the air-snow peak appears at about 1.55 m at both frequencies. The relative power is also much lower because the snow scatters light in more heterogeneous directions than the metal plate.

## 4 Results and Discussion

### 4.1 Altimeter "*Stare*" Mode

We start with examples of Ka- and Ku-band VV power (in dB) along both the northern and southern transect loops
(**Figure 7**) obtained on 16 January 2020. Results are shown as both the radar range from antenna (in meters) along with the VV power (in dB) along a short transect distance. Several key features are immediately apparent. For both Ka- and Ku-bands, the dominant VV backscatter tends to originate from the air/snow interface, primarily due to significant surface scattering contribution from this interface. The Ku-band signals also exhibit strong backscatter from greater ranges, which could correspond to volume scattering in the snow, layers with different dielectric properties caused by density
inhomogeneities, and/or the snow/sea ice interface. The key difference between the Ka- and Ku-bands is that, owing to the shorter wavelength of Ka-band, the attenuation in the snow pack is larger. Thus, compared to Ku-band, the dominant return from Ka-band is expected to be limited to the air/snow interface, while Ku-band penetrates further down through the snow volume and scatters at the snow/sea ice interface. In other words, the extinction (scattering + attenuation) in the snow in Ka-band is higher than Ku-band, and therefore, the snow/sea ice interface is hard to detect using Ka-band. Note that the power
that comes from above the air/snow interface within a few cm of the peak is the impulse response of the radar. The noisy power at the -60 dB level is probably a range sidelobe of the signal from the peak region. All FMCW radars have range sidelobes, which are due to the non-ideal behaviour of the instrument as well as artefacts of the Fourier transform of a windowed signal. If the radar introduces no distortions, there will be a first sidelobe at a level of -32 dBc and a second sidelobe at a level of -42 dBc (dBc being relative to the peak).

In this example, the local peak at the air/snow interface is generally stronger in the Ku-band than the local peak at the snow/ice interface, but this will depend strongly on the geophysical and thermodynamic state of the snow pack, including scatterer size, snow depth, density and composition (wind slab or metamorphic snow), snow salinity and temperature (if the snow pack is saline). Instances along the transect where the backscatter is greater at depth are apparent. **Figure 7** also highlights the influence of snow depth on the backscatter, with less penetration and less multiple scattering observed for the
data collected along the southern transect, which consisted of a mixture of FYI in refrozen melt-ponds and intermittent SYI with overall shallower snowpack. For the northern transect, the cross-polarized correlation coefficient (and indicator of the strength of multiple scattering) shows that multiple scattering is dominating from a depth below 1.8 m in the Ka-band, and



from a depth below 2.2 m in the Ku-band (not shown). There is considerably less multiple scattering in the southern transect data. However, further research is necessary to determine which type of multiple scattering (e.g. volume/surface, surface/surface, or volume/volume) is dominant from the signal contributions; and is beyond the scope of this paper.


**Figure 8** shows the average of the range profile of VV- and HH-polarized signal power for the same date/time as in **Figure 7**, yet processed for two different locations along the same transect segment (see figure caption). The range displayed is limited to 3.0 m and the data are zoomed in sections of 6 m width (6 m of travel along the transect). Only independent samples are included, where the speed of the sled is at least 0.4 m/s. In **Figure 8(a)**, both Ku- and Ka-bands have a peak return between 1.5 and 1.6 m range, with peak HH backscatter of -20.8 and -30.2 dB, respectively (VV backscatter is similar at -20.6 and -29.7 dB). Power is also returned in the Ku-band at a range of approximately 2.0 m. This could be either a strong return from the snow/ice interface or from ice layers/highly dense wind slab within the snowpack. The shallow slope of the tail of the Ku-band waveform suggests volume scattering and/or multiple scattering from the upper layers of the snow volume, whereas the tail falls off faster for Ka-band.


**Figure 8(b)** is an example further along the transect; at Ku-band, there are 3 peaks corresponding to ranges between 1.5 and 1.75 m (first peak at 1.52 m, second and third peaks at 1.66 and 1.73 m, respectively). There is also power returned from 1.94 m. This peak is 42 cm below the first peak, which could correspond to the snow/ice interface. Snow depths from MagnaProbe ranged from a shallow 7 cm to as deep as 53 cm, with a mean depth of 23 cm (median of 19 cm). Note however, that the peak separations stated here assume the relative dielectric constant is 1.0. Given the bulk snow densities, ranging from 256.5 to 312.6 kg m$^{-3}$, wave propagation speed was calculated to be around 80% of the speed in a vacuum. Therefore, the separation between peaks at greater range than the air/snow interface is around 80% of what it appears to be in the data as shown here, where all data are scaled for the speed of light in free space.


For the shallower snow cover over the southern transect shown in **Figure 8** at 26 – 31 m **(c)** and 150 – 156 m **(d)**, there is less multiple scattering within the snow and the long tail falls off faster. In the examples shown, the dominant backscatter at both Ka- and Ku-bands comes from the air/snow interface, with Ku-band and Ka-band in **Figure 8(d)** also picking up a secondary peak between 1.6 and 1.8 m, which could correspond to the snow/sea ice interface. The MagnaProbe data along this portion of the transect had mean and median snow depths of 13 and 11 cm, respectively.


These VV (and HH) data demonstrate the potential for detailed comparisons between KuKa data and coincident datasets such as snow MagnaProbe and SMP to explore the scattering characteristics in the Ka- and Ku-bands, over varying snow and ice conditions. Further insight is gained by overlaying the MagnaProbe snow depth (**Figure 9** for a section of the northern transect). Also shown is the first peak identified using a simple peak detection method that corresponds to the snow/air interface. Of note is that there appears to be agreement between the first peaks detected in the Ka- and Ku-bands, and between peaks in the Ku-band echoes and the MagnaProbe snow depths (which have been scaled by 0.8 to take into considering the slower wave propagation speed into the snow). The mechanisms whereby the $\sigma_{VV}^0$ increases at the snow/ice interface, and correlations between snow depth and these peaks, will be further investigated. Further, the vertical resolution







of the instrument is sufficient to resolve features within the snow pack such that scattering surfaces can be identified and their relative contributions to the backscatter investigated.

Finally, we show the example of backscatter from the highly-saline, refrozen lead covered by frost flowers sampled on 24 January 2020 when the ice was approximately 10 cm thick (**Figure 10**). As expected, there is a strong backscatter return
from the rough effective air/sea ice interface surface produced by brine wicking in the frost flowers at both Ka- and Ku-bands, with little scattering below the lead surface. Coincident to the radar measurements, we also measured frost flower and ice salinities at 1 cm resolutions. The top 1 cm salinity was ~ 36 ppt, and the bulk ice salinity was ~ 10 ppt (not shown). These high salinities are expected to mask the propagation of Ka- and Ku-bands signals to reach the ice/water interface.

**4.2 Scatterometer "*Scan*" Mode**

The observed hourly-averaged Ka- and Ku-band $\sigma^0_{VV}$, $\sigma^0_{HH}$ and $\sigma^0_{HV}$ and derived polarimetric parameters $\gamma_{CO}$, $\gamma_{CROSS}$, $\varphi_{VVHH}$ and $\rho_{VVHH}$ from the snow-covered SYI, acquired on 10 and 15 November 2019 are presented in Figures 10(a) to (e), to illustrate the polarimetric backscatter and parameter variability, as a function of $\theta$. Errors bars for the Ka- and Ku-band $\sigma^0_{VV}$, $\sigma^0_{HH}$ and $\sigma^0_{HV}$ are displayed as standard deviation of the backscatter, as a function of incidence angle, throughout the hourly scans. The standard deviation of the $\gamma_{CO}$, $\gamma_{CROSS}$ and $\varphi_{VVHH}$ are estimated from the probability density functions of
these parameters, following Geldsetzer et al. (2007) and Lee et al. (1994), while variability in $\rho_{VVHH}$ are displayed as minimum-maximum range.

**4.2.1 Ka- and Ku-band $\sigma^0_{VV}$, $\sigma^0_{HH}$ and $\sigma^0_{HV}$**

**Figure 11 (a)** and **(b)** illustrate Ka- and Ku-band $\sigma^0_{VV}$, $\sigma^0_{HH}$ and $\sigma^0_{HV}$ signatures from a homogenous 12-cm snow-covered refrozen melt-ponded SYI, acquired on 10 and 15 November 2019, as air (near-surface) temperature increased from -28°C (-
35°C) (10 November) to -12°C (-12°C) (15 November), measured from the ship (**Figure 3**) and the RS site-installed DTC (**Figure 4(a), (b)**), respectively. The increase in air and near-surface temperature between 10 and 15 November occurred during a minor storm event with ~ 15 m/s windspeed and corresponding snow redistribution. Between 10 and 15 November, our results demonstrate an increase in Ka- and Ku-band $\sigma^0_{VV}$ and $\sigma^0_{HH}$ by ~ 6 dB and ~ 3 dB, respectively. The steep increase in backscatter is prominent at nadir- to near-range $\theta$ ~ 5° (Ka-band) and ~ 10° (Ku-band). Variability and increase in nadir-
and near-range backscatter can be attributed to either increase in surface scattering (denser or smoother snow surface or smoother ice surface at nadir), or volume scattering (larger snow grains), also potentially leading to variations in Ku- and Ka-band radar penetration depth between the cold and the warm day. Temperature, influencing snow metamorphosis (snow grain growth) and changes in dry snow properties like surface roughness, e.g. from erosion, deposition, or wind compaction can result in increased backscatter within the scatterometer footprint. Snow surface temperatures from the radar footprint
measured from the TIR camera (installed next to the radar system) recorded an increase in the snow surface temperatures from ~ -28°C (10 November) to ~ -8°C (15 November) (**Figure 4(c)**). These changes observed from the TIR camera are consistent with the near-surface and snow surface temperatures measured by the DTC, installed next to the RS site (**Figure 4(a), (b)**).





Overall, the co-polarized backscatter magnitude is higher at nadir and near-range $\theta$, for both Ka- and Ku-bands, and
demonstrates a steady decline at mid- and far-range $\theta$, especially for Ku-band. However, at $\theta > 35°$, Ka-band $\sigma_{VV}^0$ and $\sigma_{HH}^0$
shows a characteristic increase by $\sim$ 3 dB (15 November) and 5 dB (10 November), likely due to strong volume scattering
from the topmost snow surface, with the footprint covered at far-range $\theta$ likely to be spatially less-homogenous. However,
more analysis using snow/sea ice geophysical properties, including snow redistribution and surface roughness changes; and
meteorological conditions, is required in this regard, and is outside the scope of this paper. The error for the co-polarized
backscatter ranges between ±2.1 dB (Ka-band) and ±1.9 dB (Ku-band) at nadir- and near-range $\theta$, and decreases to ±2.0 dB
(Ka-band) and ± 1.7 dB (Ku-band) at mid- and far-range $\theta$. As expected, at near-range $\theta$, the error is dominated by low
number of independent samples, while the signal-to-noise ratio reduces with increase in the number of independent samples,
at mid- and far-range $\theta$. These ranges are consistent for measurements acquired during the cold and warm periods on 10 and
15 November, respectively.

In the case of cross-polarized backscatter $\sigma_{HV}^0$, Ka-band backscatter is dominant throughout the $\theta$ range, with an $\sim$ 10 dB
increase in $\sigma_{HV}^0$, compared to Ku-band $\sigma_{HV}^0$, on both 10 and 15 November. This substantial increase in Ka-band $\sigma_{HV}^0$ indicates
strong volume scattering contribution from the topmost snow layers, compared to lower Ku-band volume scattering from
within the penetrable snow volume within the snow pack. For both Ka- and Ku-bands, overall, the $\theta$ dependence on $\sigma_{HV}^0$ is
mostly negative, with both frequencies exhibiting a steady decline with $\theta$. Although, Ku-band dependence is slightly more
negative than Ka-band at near-range $\theta$, followed by a slight increase in the mid-range, and followed by slightly negative
dependence at far-range $\theta$. In addition, the signal-to-noise ratio of both Ka- and Ku-band $\sigma_{HV}^0$ is comparable and consistent
to the $\sigma_{VV}^0$ and $\sigma_{HH}^0$ signal-to-noise ratio. Between Ka- and Ku-band $\sigma_{HV}^0$ signatures from 10 and 15 November, both
frequencies demonstrate only an $\sim$ 2 dB difference, consistently throughout the $\theta$ range. Detailed analysis of all the
polarimetric backscatter signatures from both frequencies are outside the scope of this paper.

**4.2.2 Ka- and Ku-band $\gamma_{CO}$, $\gamma_{CROSS}$, $\varphi_{VVHH}$ and $\rho_{VVHH}$**

The co-polarized ratio $\gamma_{CO}$ demonstrates little difference between $\sigma_{VV}^0$ and $\sigma_{HH}^0$ for both Ka- and Ku-bands, for both 10
and 15 November observations (**Figure 11(c)**). At $\theta > 20°$, Ku-band $\gamma_{CO}$ illustrates a slightly higher magnitude at $\sigma_{VV}^0$ over
$\sigma_{HH}^0$. These observations are consistent with scattering models assuming isotropic random media (Lee et al., 1994), and
similarly observed from MYI observations from a C-band scatterometer system (Geldsetzer et al., 2007). The cross-
polarized ratio $\gamma_{CROSS}$ shows characteristic shift in Ka-band when compared to Ku-band, especially at nadir- to 5°, where
Ka-band $\sigma_{HH}^0$ dominates over $\sigma_{HV}^0$ on 15 November (**Figure 11(d)**). This suggests strong surface scattering from the denser
or smoother snow surface or smoother ice surface at nadir. With increasing $\theta$, the Ka-band $\gamma_{CROSS}$ demonstrates greater $\sigma_{HV}^0$
suggesting potential volume scattering from the upper layers of the snow pack, on both 10 and 15 November. Ku-band
$\gamma_{CROSS}$ demonstrates the same behaviour like Ka-band till $\theta = 15°$, after which the cross-pol ratio remains unchanged on
both cold and warm day. The co-polarized phase difference $\varphi_{VVHH}$ for both Ka- and Ku-bands clearly demonstrate





variability in phase shifts between the cold and warm days, especially at mid- and far-range $\theta$ (**Figure 11(e)**). The higher Ka-band frequency decorrelates and undergoes higher positive phase shifts, deviating from zero, compared to the lower frequency Ku-band on both 10 and 15 November. This suggests significant Ka-band anisotropy from the snow surface between the cold and warm day, while the lower phase difference at Ku-band indicates isotropic scattering, possibly from

randomly distributed, non-spherical scatterers (Nghiem et al., 1990; Nghiem et al., 1995; Drinkwater et al., 1995b). Also note the large shift of Ka-band $\varphi_{VVHH}$ towards positive values, at $\theta > 20°$ on 15 November, and indicates potential of second- or multiple-order scattering within the snow pack, likely caused by surface roughness changes. This characteristic is less prominent from the Ku-band $\varphi_{VVHH}$. The complex co-polarized correlation coefficient $\rho_{VVHH}$ values are closer to 1 for both Ka- and Ku-bands, at nadir- and near-range $\theta$, on both 10 and 15 November (**Figure 11(f)**). The $\rho_{VVHH}$ values from 15

November are slightly higher than from 10 November, suggesting increased Ka- and Ku-band surface scattering at these angles during the warm day. Similar to the polarimetric backscatter signatures, detailed analysis of polarimetric parameters is beyond the scope of this paper.

Overall, the KuKa radar system operating in the scatterometer mode is able to characterize changes in polarimetric backscatter and derived parameters, following variations in meteorological and snow geophysical changes during a snow

warming event in the middle of winter thermodynamic regime. Prominent changes in Ku- and Ka-band backscatter and derived parameters are observed at nadir and near-range incidence angles, exemplifying its importance towards snow/sea ice state variables from satellite radar altimetry. In a warming Arctic, with potential warming and storm events occurring within the winter regime, the surface-based KuKa radar was sensitive to geophysical changes on snow-covered sea ice. This also means both frequencies may potentially exhibit varying penetration depths between the cold and warm days, influencing the

accuracy of satellite-derived snow depth retrievals from dual-frequency approaches. On the other hand, changes in backscatter and parameters throughout the incidence angle range provides the first-hand baseline knowledge of Ku- and Ka-band backscatter behaviour from snow-covered sea ice and its associated sensitivity to changes in snow/sea ice geophysical and thermodynamic properties. This is important to be applied on future Ku- and Ka-band satellite SAR and scatterometer missions for accurately retrieving critical snow/sea ice state variables, such as sea ice freeze- and melt-onset timings, or sea

ice type classification.

## 5    Conclusions

Satellite remote sensing is the only way to observe long-term pan-Arctic sea ice changes. Yet satellites do not directly measure geophysical variables of interest and therefore require comprehensive understanding on how electromagnetic energy interacts within a specific medium, such as snow and sea ice. During the MOSAiC expedition, we had the unique

opportunity to deploy a surface-based, fully-polarimetric, Ku- and Ka-band dual-frequency radar system (KuKa radar), together with detailed characterization of snow, ice and atmospheric properties, to improve our understanding of how radar backscatter at these two frequencies varies over a full annual cycle of sea ice growth, formation and decay. We were also





During the autumn (Leg 1) and winter (Leg 2) of the MOSAiC drift experiment, the instrument sampled refrozen leads, first-year and second-year ice types and refrozen melt ponds. This data thus provides a unique opportunity to characterize the autumn to winter evolution of the snowpack and its impact on radar backscatter and radar penetration, including the evolution of brine-wetting on snow-covered first-year ice, providing a benchmark dataset for quantifying error propagation in sea ice thickness retrievals from airborne- and satellite-borne radar sensors. Our observations from the transect

measurements over second-year ice illustrate the potential of the dual-frequency approach to estimate snow depth on second-year sea ice, under cold and dry (non-saline) snow geophysical conditions, during the winter season. On thin ice and first-year ice conditions, with thin and saline snow covers, our initial assessments show distinct differences in radar scattering horizon at both Ka- and Ku-band frequencies. Detailed analysis, combining snow pit and magnaprobe data to all the transect data collected is outside the scope of the present paper, and will form the basis of future work. In particular, future analyses

will focus on comparisons between the KuKa radar data and simulations, driven by *in situ* snow/sea ice geophysical properties and meteorological observations, in order to attribute the peaks and volume scattering to physical surfaces and volumes. Data to be collected during the melt-onset and freeze-up is forthcoming and should shed further insights into radar scattering horizon variability during these critical transitions.

         The dual-frequency KuKa system also illustrates the sensitivity in polarimetric backscatter and derived parameters, to

changes in snow geophysical properties (example from 10 and 15 November observations used in this study). For the first time, the radar system was able to characterize prominent changes in Ku- and Ka-band radar signatures between cold (10 November) and warm (15 November) periods, especially at nadir incidence angle; exemplifying the impact of accurate snow/sea ice state variable retrievals (e.g. snow depth) from satellite radar altimetry. Through illustrating changes in Ku- and Ka-band polarimetric backscatter and derived parameters between the cold and warm period, the dual-frequency approach

shows promise to characterize frequency-dependent temporal changes in polarimetric backscatter from snow-covered sea ice, as a function of incidence angle; applicable for future Ku- and Ka-band satellite SAR and scatterometer missions. By utilizing frequency-dependent polarimetric parameter index such as '*Dual-frequency ratio*' developed by Nandan et al. (2017c), the KuKa system will be able to reveal characteristic temporal changes in polarimetric backscatter, as a function of snow depth and sea ice type, polarization, frequency and incidence angle, as snow/sea ice system thermodynamically

evolves between freeze-up to spring melt-onset.

         Moving forward, new space borne Ku- and Ka-band radar altimeter and SAR satellites such as the ESA's CRISTAL and CSA's REM-Cryo missions (to name a few) are proposed to be launched in the near future. While the signals received from a satellite altimeter are in the far field of the antenna, whereas the signals from the KuKa radar are in the near field, the *in situ* based radar system can provide important insights into the interaction of the radar signals with the range of physically different surfaces encountered on sea ice floes. Our findings from this study, and forthcoming papers will facilitate




significant improvements in already existing Ku- and Ka-band dual-frequency algorithms to accurately retrieve snow depth and sea ice thickness from these above mentioned satellites. Datasets acquired from these forthcoming satellites will also provide a valuable source for downscaling surface-based estimates of snow depth on sea ice from the KuKa system to '*satellite scale*' and validate new or similar existing findings.

**Author contribution:** Stroeve conceptualized the design of KuKa, acquired funding for the MOSAiC expedition and the building of the KuKa Radar, participated in the data collection during leg 2, performed analysis of transect data and wrote the manuscript. Nandan participated in the MOSAiC leg 2 expedition, processed and performed analysis of the scatterometer mode data and provided review and editing. Willatt helped with data visualization, data processing and review and editing. Mead designed and built the KuKa radar, provided software for data processing and provided review and editing. Tonboe,

Huntemann, Hendricks, Ricker, Itkin, Schneebeli helped with data collection, review and editing. Krampe, Matero and Hoppmann helped with data collection. Wilkinson and Tsamados are Co-Is on NERC grant that funded the work.

**Acknowledgements**

This work was funded in part through NERC grant # NE/S002510/1, the Canada 150 Chair Program and the European Space
Agency PO 5001027396. Data used in this manuscript was produced as part of the international Multidisciplinary drifting Observatory for the Study of Arctic Climate (MOSAiC): MOSAiC20192020, AWI_PS122_00. Data are available at UK Polar Data Centre. The authors thank Marine Environmental Observation, Prediction and Response Network (MEOPAR) Postdoctoral Fellowship grant to Vishnu Nandan. The authors also thank the crew of *R/V Polarstern* and all scientific members of the MOSAiC expedition for their support in field logistics and field data collection.


**Table 1**. Summary of Ka- and Ku-band specifications.

|  | Ku-band | Ka-band |
|---|---|---|
| **Radar Parameter** | **Value** | **Value** |
| RF output frequency | 12-18 GHz | 30-40 GHz |
| Transmit power (at the output of RF unit bulkhead connector) | 10 dBm | 6 dBm |
| Transmit bandwidth | 6 GHz | 10 GHz |
| Range resolution | 2.5 cm | 1.5 cm |
| Antenna 6-dB two-way beamwidth | 16.9$^\circ$ at 13.575 GHz | 11.9$^\circ$ at 35 GHz |
| Cross-polarization isolation | >30 dB | >30 dB |
| Transmit/receive polarization | VV, HH, HV, VH | VV, HH, HV, VH |
| Chirp length | 1-99 ms (set to 2ms for normal operation) | 1-99 ms (set to 2ms for normal operation) |





| Digitizer | 14 bits resolution, 5MS/s raw sample rate | 14 bits resolution, 5MS/s raw sample rate |
| --- | --- | --- |


**Table 2.** Summary of snow pit properties along northern transect. Values are given as averages, standard deviations and min/max (in parenthesis) from 2 to 6 snow pits. Results show considerable variability in snow water equivalent (SWE) and snow depth.

| Date | Mean Snow Water Equivalent (SWE) (mm) | Mean Snow Depth (cm) | Mean Density (kg/m$^3$) | Mean Bulk Salinity (ppt) |
| --- | --- | --- | --- | --- |
| 19 December 2019 | 50.75 $\pm$ 38.07 (19,105) | 19.0 $\pm$ 12.99 (9,38) | 256.5 $\pm$ 39.46 (211.1,300.0) | 0.1 $\pm$ 0.05 (0,0.1) |
| 26 December 2019 | 36.75 $\pm$ 30.89 (14,80) | 11.13 $\pm$ 4.38 (6,16) | 312.6 $\pm$ 206.99 (147.4,615.4) | 0.1 $\pm$ 0.14 (0,0.3) |
| 2 January 2020 | 44.75 $\pm$ 36.25 (15,96) | 16.13 $\pm$ 12.69 (8,35) | 270.3 $\pm$ 74.10 (187.5,366.7) | 0.2 $\pm$ 0.21 (0,0.5) |
| 9 January 2020 | 53.25 $\pm$ 29.39 (26,88) | 19.75 $\pm$ 9.03 (12,32) | 261.6 $\pm$ 55.57 (185.7,319.0) | 0.0 $\pm$ 0.05 (0,0.1) |
| 16 January 2020 | 71.0 $\pm$ 39.23 (31,125) | 24.0 $\pm$ 11.19 (14,40) | 286.3 $\pm$ 46.16 (221.4,325.0) | 1.8 $\pm$ 2.40 (0.1,3.5) |
| 20 January 2020 | 57.4 $\pm$ 33.19 (25,105) | 19.8 $\pm$ 11.78 (9,38) | 288.8 $\pm$ 20.01 (270.0,315.8) | 0.1 $\pm$ 0.21 (0,0.6) |


**Table 3.** Dates for when the northern and southern transects were conducted, in addition to dates when the instrument sampled lead/frost flowers as well as first-year ice at the runway site.

| Date | Northern Transect | Southern Transect | Lead/Frost Flowers | Runway – first-year ice |
| --- | --- | --- | --- | --- |
| 7 November 2019 | X | | | |
| 14 November 2019 | X | | | |
| 23 November 2019 | | | X | |
| 20 December 2019 | X | | | |
| 26 December 2019 | X | X | | |





| Date | | | | |
|---|---|---|---|---|
| 2 January 2020 | X | X | | |
| 9 January 2020 | X | X | | |
| 12 January 2020 | | | | X |
| 16 January 2020 | X | X | | |
| 19 January 2020 | | | | X |
| 23 January 2020 | | | X | |
| 24 January 2020 | | | X | |
| 30 January 2020 | X | X | | |

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





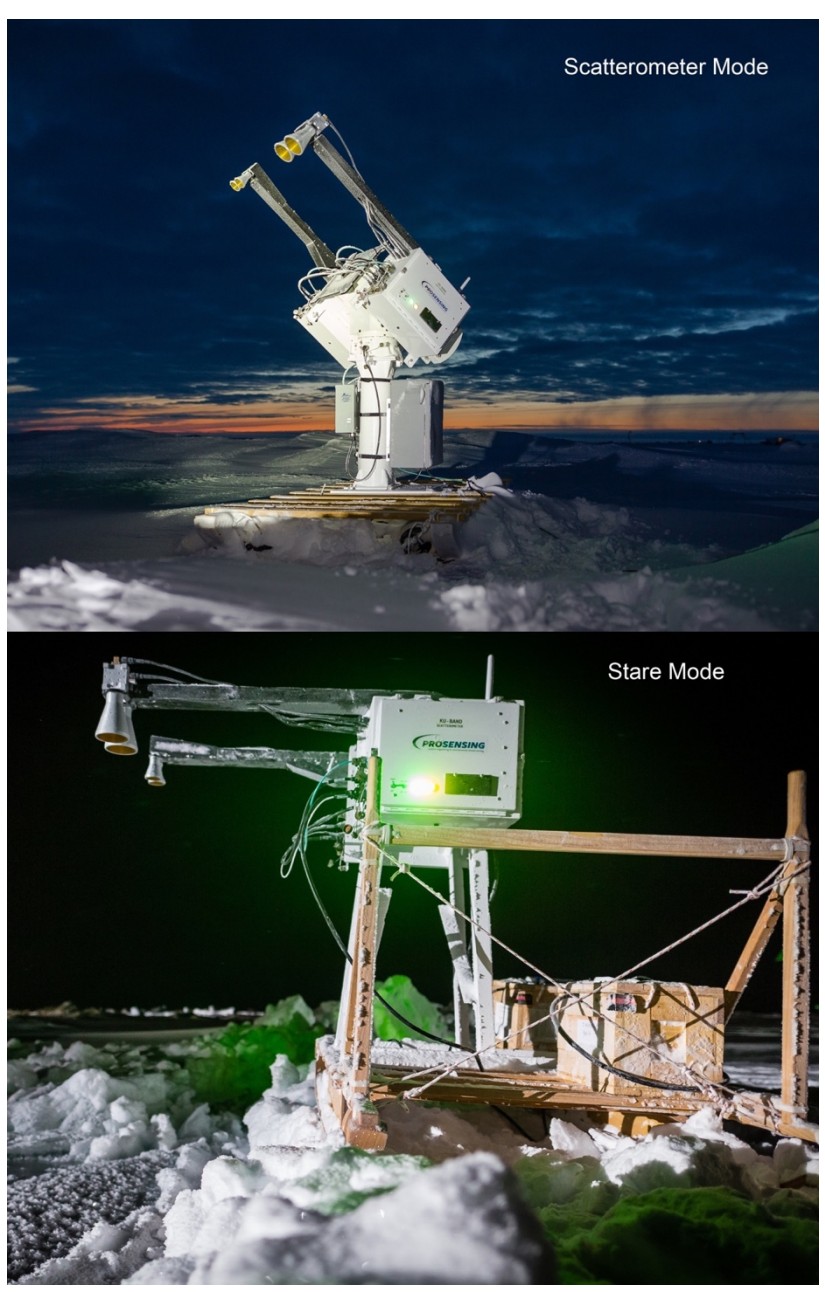


**Figure 1.** Configuration of KuKa radar in scatterometer "*scan*" (top) and altimeter "*stare*" (bottom) modes. Photo Credit: Stefan Hendricks.





**Figure 2.** Annotated schematic of the Central Observatory (CO) around *R/V Polarstern*. The schematic is overlaid on a post-processed airborne laser scanner map, acquired on 21 February 2020. The remote sensing site is denoted by 'RS'.



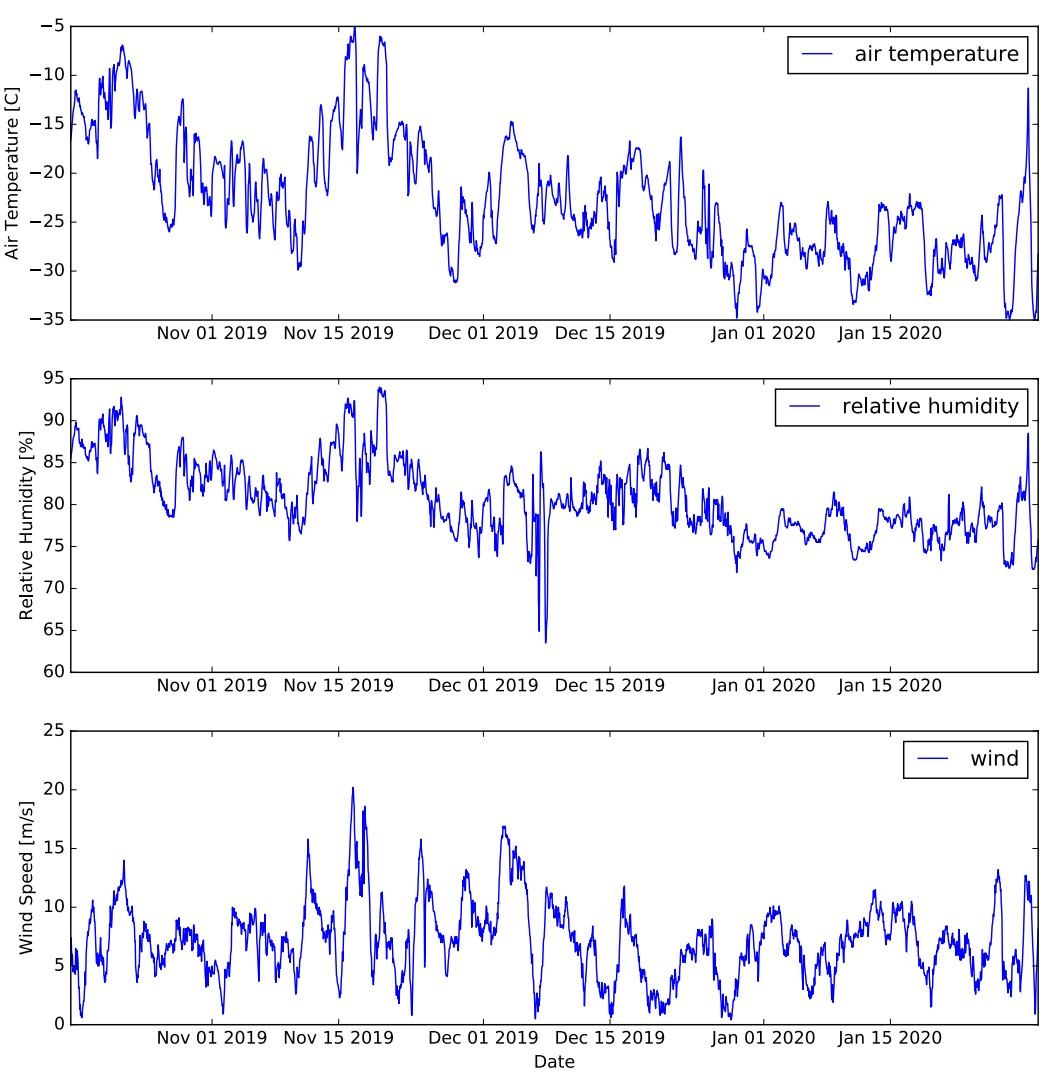

**Figure 3.** Summary of weather data during deployment of KuKa radar, measured from *R/V Polarstern*. Shown are the air temperature, relative humidity and the wind speed from 18 October 2019 to 31 January 2020 at 30 m height.





**Figure 4.** Hourly-averaged near-surface, snow and sea ice temperature gradient from the RS site, acquired by thermistor
strings on (a) 10 and (b) 15 November 2019. The top 20 cm represents the distance between the first temperature sensor
located above the air/snow interface and the temperature sensor located at the air/snow interface. The bright yellow pixels
represent the snow volume. The thermistor string was installed on 7 November 2019. (c) Hourly-averaged snow surface
temperature from the RS site between 10 and 15 November 2019, acquired by the TIR camera.





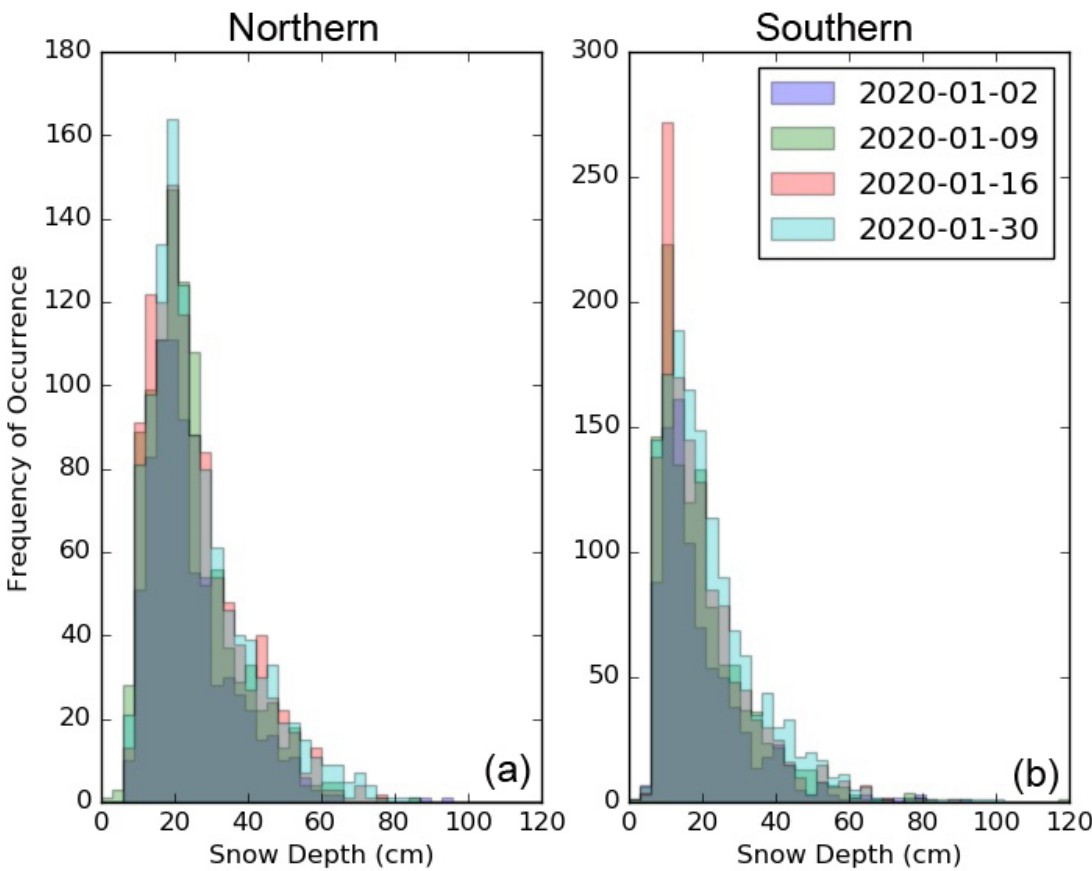

**Figure 5.** Snow depth distribution during January 2020 along the Northern (a) and Southern (b) transect loops.





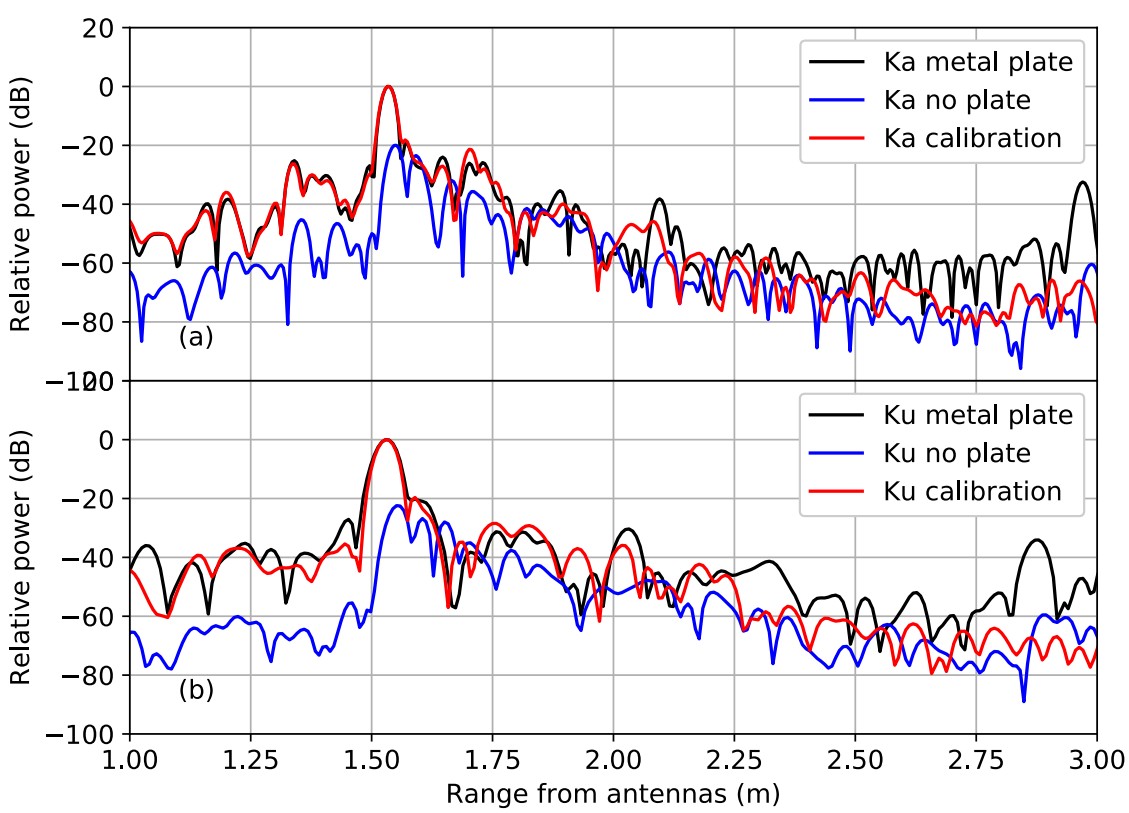

**Figure 6.** Radar returned power in the (a) Ka- and (b) Ku-bands. These data were gathered over the exposed snow and ice (blue), a metal plate on the snow surface, approximately 15 x 55 cm (black) and the internal calibration loop (red). The calibration data have been shifted in range and power to correspond to the peak locations of the metal plate. The power that comes from above the air/snow interface within a few cm of the peak is simply the impulse response of the radar. The noisy power at the -60 dB level is probably a range sidelobe of the signal from the peak region. The range sidelobes at the -23 dB level and below (Ka-band) -30 dB level and below (Ku-band) and are due to internal reflections in the radar.



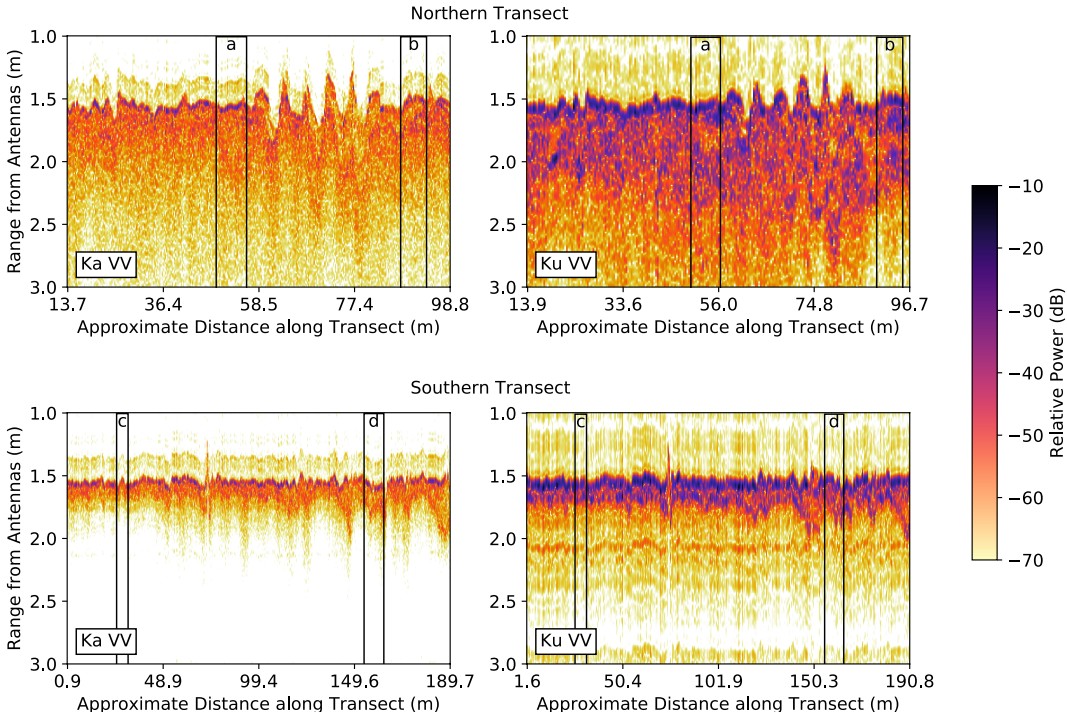

**Figure 7.** Ka- (left) and Ku-band (right) VV-polarized power as a function of distance along the northern (top) and southern transect (bottom). Data acquired on 16 January 2020 at 10:52 and 12:02 UTC for the northern and southern transects, respectively. Letters a-d denote four sections shown in more detail in Figure 7, each 6 m wide (corresponding to 6 m of travel along the transect). Data are not evenly spaced along the x-axis; tick marks indicate distances along the transect where the samples were obtained.



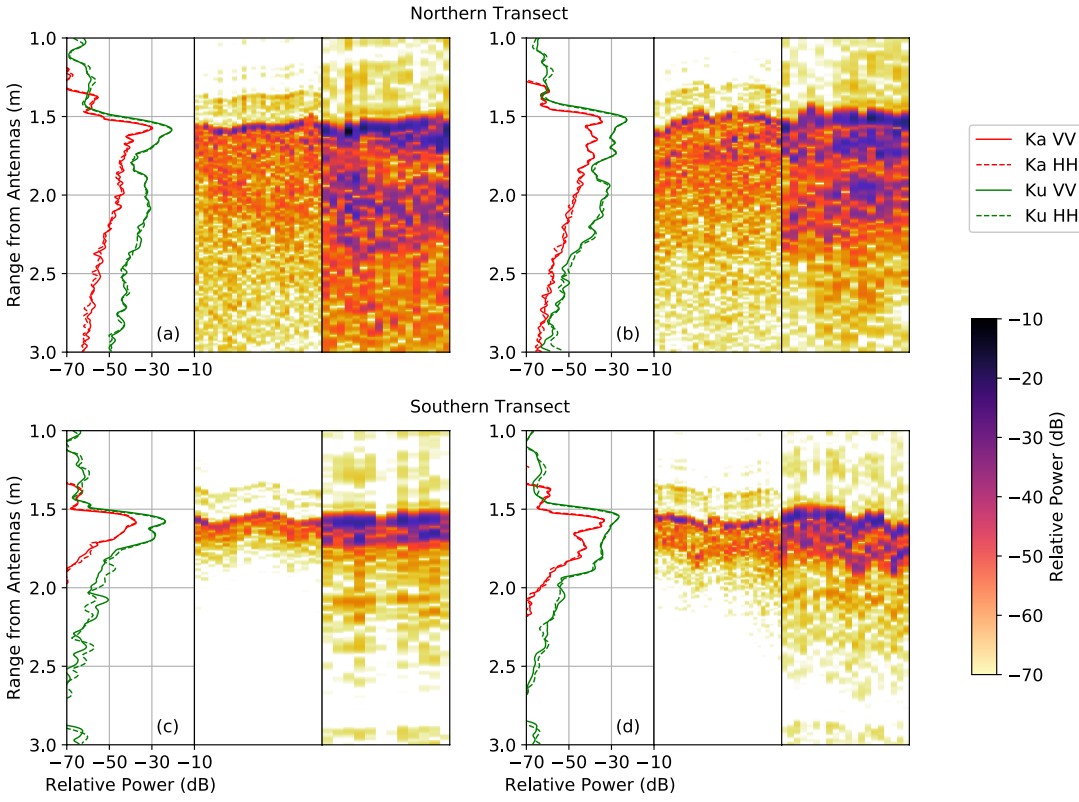

**Figure 8.** Average VV- and HH-polarized signal power as a function of range at Ku- and Ka-band for specific locations along the northern (a,b) and southern (c,d) transects as shown in Figure 7. The difference in the average spectrum between (a,b) and (c,d) is that they are from different locations along the transect and highlight the influence of multiple scattering in the snow and a return from what could be the snow/ice interface at Ku-band.





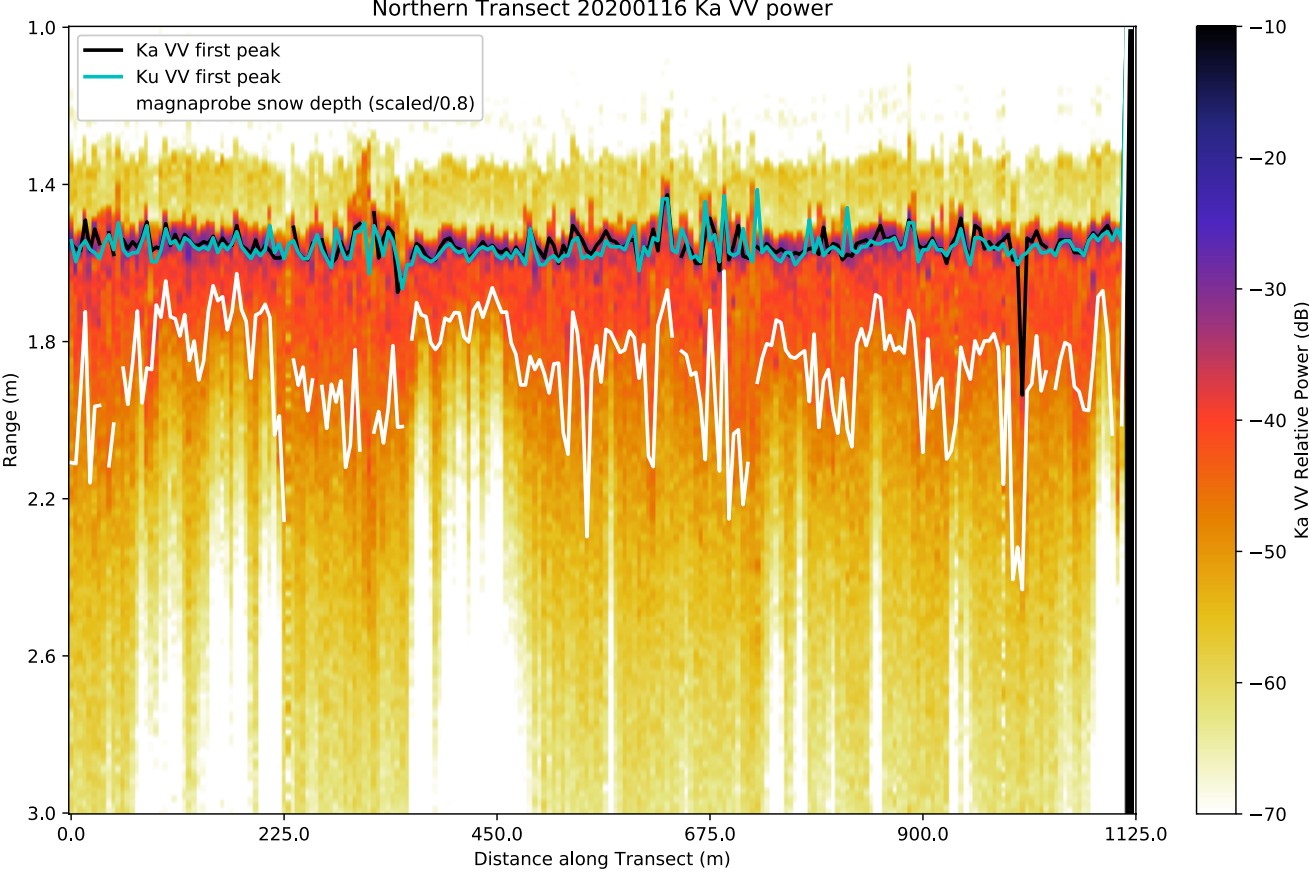

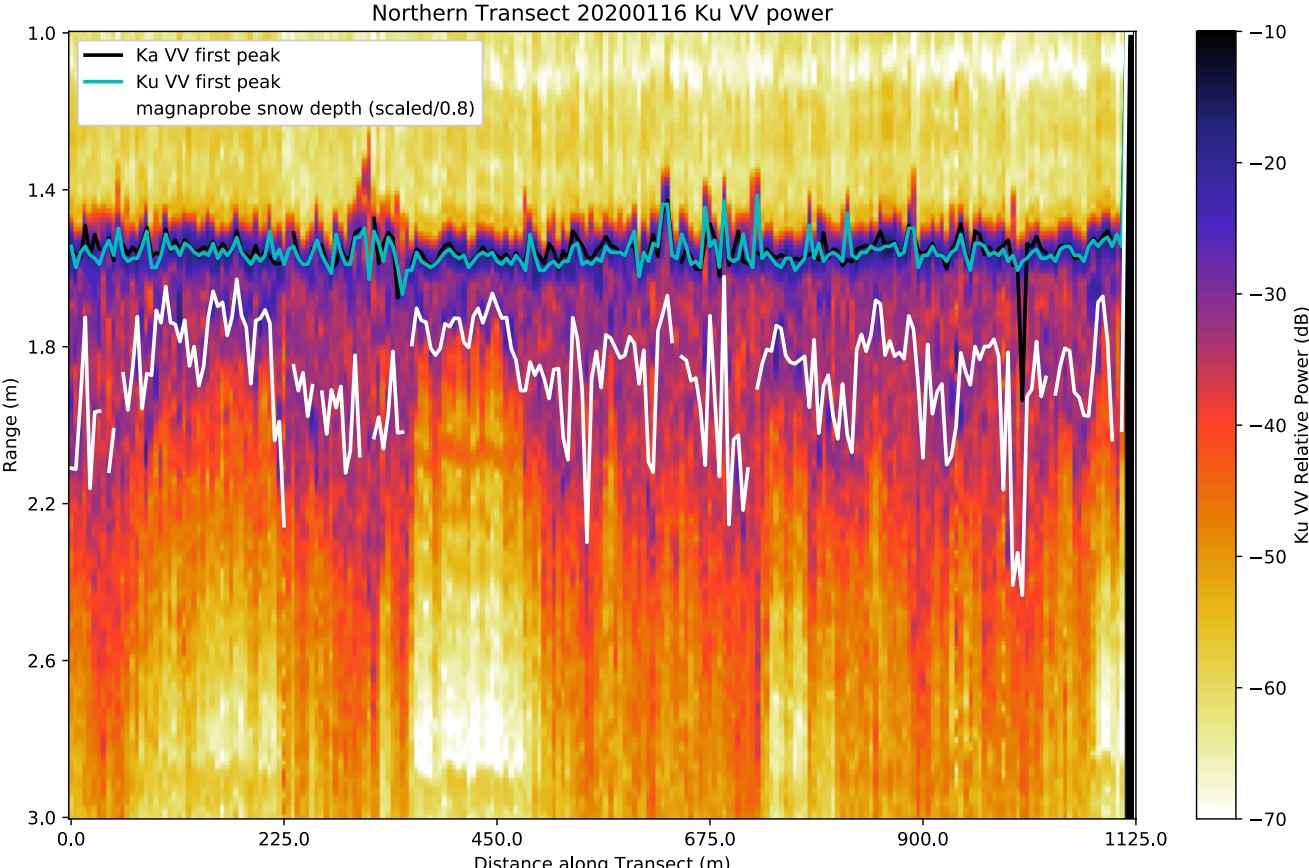

**Figure 9.** Ka (top) and Ku (bottom) VV power along the Northern Transect on 16[th] January 2020. The black and cyan lines
indicate the ranges of the first peaks detected in the Ka and Ku echoes, respectively. The white line indicates the snow depth
(from nearby magnaprobe data) plotted with depths measured from the Ka VV first peak for each echo and divided by 0.8
for comparison with the radar data, to account for the slower EM radiation propagation of the radar in snow relative to free
space.


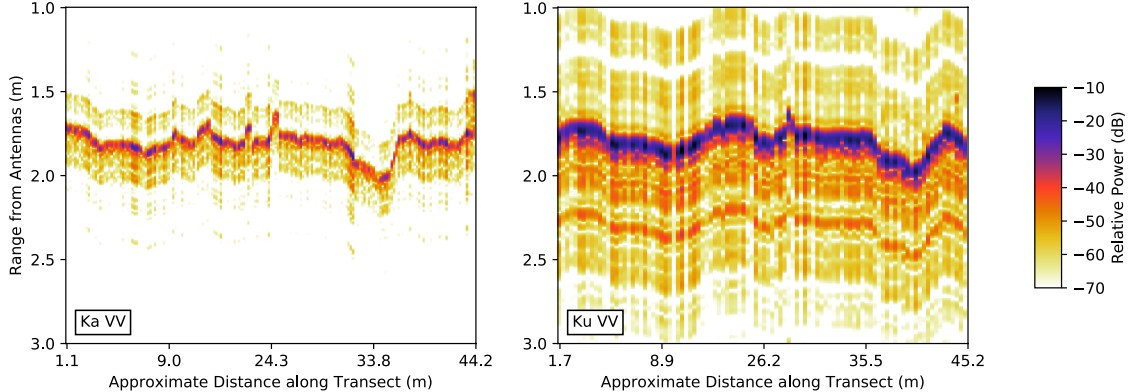

**Figure 10**. Ka- (a) and Ku-band (b,) VV-polarized signal power as a function of distance along the refrozen lead. Data
acquired on 24 January 2020 at 12:41 UTC.








**Figure 11.** Ku- and Ka-band polarimetric backscatter and parameters from snow covered sea ice from the RS site acquired on 10 November (Cold) and 15 November 2019 (Warm). (a) Ka-band Co- and cross-polarized backscatter $\sigma_{VV}^0$, $\sigma_{HH}^0$ and $\sigma_{HV}^0$; (b) Ku-band Co- and cross-polarized backscatter $\sigma_{VV}^0$, $\sigma_{HH}^0$ and $\sigma_{HV}^0$ (c) Co-polarized ratio $\gamma_{CO}$; (d) Cross-polarized ratio $\gamma_{CROSS}$; (e) Co-polarized phase difference $\varphi_{VVHH}$; and (f) Co-polarized correlation coefficient $\rho_{VVHH}$. Fit lines are cubic for
backscatter and error bars represents standard deviation. Fit lines for co-pol ratio, cross-polarized ratio and co-polarized correlation coefficient are quadratic. Errors bars for these parameters represent standard deviation (co-polarized and cross-



polarized ratio) and min-max (co-polarized correlation coefficient). Error bars for co-polarized phase difference represent standard deviation.
