# Peer review of "Surface-Based Ku- and Ka-band Polarimetric Radar for Sea Ice Studies"

_The Cryosphere, 2020_

## Referee Comment (RC1) · Nathan Kurtz (Referee) · 9 Sep 2020

Nathan Kurtz (Referee)

nathan.t.kurtz@nasa.gov

This is a well written paper which describes in-situ measurements from a Ku/Ka band radar during a portion of the MOSAiC field campaign. The results are highly relevant and represent a very good data set needed for better understanding of radar returns from sea ice and the snow layer. From a technical perspective I believe the methods and data are quite sound and thoroughly described in the paper. However, the paper stops short of providing information that would be of most use as a reference to understanding or improving retrieval techniques from satellite or airborne altimeters. What would be most useful here would be some results and discussion showing the observed difference in backscatter between the snow-air and snow-ice interfaces as well as the variability that was seen. Tying these to some of the physical properties

measured in-situ such as salinity would also be useful. It is mentioned that this paper represents a first data set and that further studies will follow, from that perspective I do think the paper is indeed quite useful and should be published. But it would be nice to have the results placed in a bit better context towards how they could specifically be used in understanding satellite or airborne radar returns.

Some specific comments are below:

Line 280: The MOSAiC floe description seems a bit simplistic with declarations about the floe properties which ignore some of the variability within the floe itself. I would suggest referencing the paper by Krumpen et al. on the floe description and history. Some additional description of the ice properties beneath the radar site would also be helpful here and in other sections of the text such as Line 290 when it is brought up that the radar was moved.

Krumpen, T., Birrien, F., Kauker, F., Rackow, T., von Albedyll, L., Angelopoulos, M., Belter, H. J., Bessonov, V., Damm, E., Dethloff, K., Haapala, J., Haas, C., Harris, C., Hendricks, S., Hoelemann, J., Hoppmann, M., Kaleschke, L., Karcher, M., Kolabutin, N., Lei, R., Lenz, J., Morgenstern, A., Nicolaus, M., Nixdorf, U., Petrovsky, T., Rabe, B., Rabenstein, L., Rex, M., Ricker, R., Rohde, J., Shimanchuk, E., Singha, S., Smolyanitsky, V., Sokolov, V., Stanton, T., Timofeeva, A., Tsamados, M., and Watkins, D.: The MOSAiC ice floe: sediment-laden survivor from the Siberian shelf, The Cryosphere, 14, 2173–2187, https://doi.org/10.5194/tc-14-2173-2020, 2020.

Line 900-905: What does it mean that "the power that comes from above the air/snow interface within a few cm of the peak is simply the impulse response of the radar"? I'm not sure why the peak is shifted several cm, and if it is the impulse response of the radar then why does it look so different than the metal plate and calibration examples? Perhaps instead it reflects the surface height distribution and scattering characteristics of the surface.

Figure 7 and other figures: Does the Range from Antenna represent the range assuming a speed of light in free space or in snow?

Figure 9 and Line ~440: This figure and discussion are very interesting and is highly relevant for assumptions about the radar backscatter from the snow-air and snow-ice interfaces from altimeters. However, this is very qualitative and hard to put the results in a useful context as presented here. It would be useful to show at the least the backscatter difference between the snow-air interface from the algorithm and the value from the magnaprobe location.

---

## Referee Comment (RC2) · Anonymous Referee #2 · 21 Sep 2020

Review of 'Surface-Based Ku- and Ka-band Polarimetric Radar for Sea Ice Studies' by Stroeve et al.

This work describes a novel ground-based dual-frequency radar used to evaluate interactions with snow on sea ice at frequencies common to CryoSat-2, AltiKa, and their follow-on mission CRISTAL. Microwave interactions with snow are discussed as a key uncertainty limiting accuracy of altimetry-based freeboard and a potential source of snow depth information leveraging frequency dependent interaction. Stationary scans and transect measurements completed as part of MOASiC highlight complex interactions and a multitude of causal mechanisms. An extensive set of snow measurements are introduced, which in the future, will be used to decompose variations in penetration and scattering. The work focuses on introducing dual-frequency theory, the KuKa

radar system, and provides case-studies to evaluate spatial, temporal, and angular dependency of the observed environment.

Overall, the work contributes highly relevant results to the remote sensing of sea ice and the datasets described will to be critical to ongoing retrieval development. The case study examples provide a good variety of configuration and observed conditions, illustrating many of the complexities involved. However, I felt the system descriptions and quantitative backscatter analysis should be strengthened prior to publication. For the system description, the reader is directed to previous literature to establish methods for calibration and processing. Given advancements of the current system (larger bandwidth, transect profiling, multiple frequencies), additional information is needed to understand system uncertainty with respect to the relative backscatter and NRCS analysis:

(1) Calibration procedures for NRCS and relative backscatter: On lines 276 and 373 corner reflector calibration is mentioned and methods are attributed to those introduced in Geldsetzer et al. (2007) and King et al. (2013). The referenced studies indicate that external targets and background noise estimates were necessary for absolute calibration of NRCS. A factor for this is applied in Eqn (3) of the supplement but it is unclear what is used to accomplish the calibration. In depth analysis of system uncertainty is beyond scope, but an indication of system stability throughout the campaign or between scan types is needed if magnitudes are to be compared across extended temporal periods or between sites.

(2) Estimation of NRCS: Figure 11 introduces estimates of backscatter as a function of incidence angle, but I was unable to determine how the integrated impulse range was defined. I would expect a dynamic approach where the ground projected footprint is elongated with range and there is frequency / spatial dependency of penetration. How was impulse range addressed in determining NRCS and was it identical for both frequencies?

[Figure]

(3) Discussion of errors: Errors for NRCS are described briefly on line 485 as being plus minus 1.7 dB or greater. What sorts of challenges might this present in analysis and are these errors stable between configurations and deployments? I am also interested in understanding if the errors described for NRCS (Figure 11) are valid for the individual or mean waveform analysis given range dependent noise demonstrated in Figure 6.

With respect to backscatter analysis, the presented interpretations could be enhanced with additional quantitative analysis. For example, Figure 9 and the associated text describes a good agreement with the snow depth, but it is unclear how this agreement was determined. Extraction of relative backscatter at the identified interfaces might be useful when associating peaks identified in the mean profiles (Figure 8). Finally, the acknowledgements section indicates that data are available at the UK Polar Data Centre but I was unable to retrieve them using common terms from the paper. If the data are available a set of links would be helpful.

Thanks to the authors for their dedicated work to develop and deploy a novel radar system in this challenging environment. The datasets and analysis are valuable contributions to community and will improve understanding snow-radar interactions on sea ice. Please find my specific comments indicated below with page and line numbers.

Specific comments P5 L139 – The seasonal aspect is an important distinction from previous ground-based Ku-band studies on sea ice. Could a table 3 be amended or a description be added to inform the reader of how the preliminary dataset fits in the context of the larger seasonal acquisitions?

P7 L218 – Questions regarding calibration: Has absolute (external) calibration of the system been completed or is the calibration process solely internal? The supplementary materials state an absolute calibration was completed and P9 L275 mentions a corner reflector, but I had difficulty determining the process or how often it is required. Please include information on how external factors (changes in cabling or mounting structure) were accounted for in the calibration process.

[Figure]

P7 L225 – Can you add a description of the physical separation between the two horns? This is needed to reproduce the overlap estimates.

P8 L244 – Given that the system is operated on drifting ice have steps have been taken to compensate for ice motion when applying velocity or displacement thresholds? Please include a statement on uncertainty associated with ice motion and properties of the GPS.

P9 L258 – On P12 L374 it is stated that a near-field correction is applied but this line states it was not necessary. Please clarify.

P9 L272 – Is the minimum threshold of $\frac{1}{2}$ the antenna diameter greater than the accuracy of the GPS?

P10 L295 – A statement on standard snow pit protocols would be a helpful addition here. I read this currently as the SMP being a primary tool. It should be noted that the SMP provides a profile of micromechanical properties and requires a statistical model to derive microstructure. Presumably, the snow castings for uCT are to be used as a high-quality reference but this is not clear.

P10 L 311: Sturm and Holmgren, 2017 can be used as a citation for the instead of the patent: Sturm, M., & Holmgren, J. (2017). An Automatic Snow Depth Probe for Field Validation Campaigns. Water Resources Research, 9695–9701. DOI: 10.1029/2018WR023559@10.1002/(ISSN)1944-7973.SNOWEX1

Figure 2 and P10 L312: I was not able to determine in Figure 2 where the North and South transects were completed. Could these be added?

Figure 5 and P11 Line 343: The snow depth distributions are shown as a frequency count. Are they exact repeats in terms of the number of observations and locations?

P12 L379: Does the internal calibration loop agreement with the metal plate suggest the system is well constrained and does not need external calibration? A clear statement on system stability would provide confidence that the measurements can be

compared between deployments and configurations. Alternatively a statement of why external calibration is not needed could be provided.

P13 L390 and P13 408 411: Can quantities be associated with the terms dominate and significant? For example, what % of the signal is coming from the AS interface at each frequency or what % of backscatter is observed at ranges beyond where the SI interface is presumed to be.

P14 L425: How are location of the peaks determined? Its hard to pick where the peaks are by eye beyond the main lobe in Figure 8.

P14 L434: Does the cross-pol ratio support the presence of increased multiple scattering for the Northern Transect?

P14 L441: Peak detection = greatest magnitude? This is a critical point of clarification as several studies have suggested that surface bedforms and/or roughness dictate where along the rising edge the AS interface can be found.

P14 P442 and Figure 9: What is used to define agreement between the Ku-band signal and snow depth? Please provide quantities to support.

P15 L463: There needs to be some explanation about how estimates of NRCS were generated. What impulse range is integrated? Is the same for both systems? Does it change with incidence angle?

P16 L510: There appears to be little or no difference at angles common to altimetry (near nadir figure 11e) between the two dates. How should this finding be interpreted for current and upcoming missions?
* * *

---

## Author Comment (AC1) · 14 Oct 2020

**Response to the Reviewer #1**

This is a well written paper which describes in-situ measurements from a Ku/Ka band radar during a portion of the MOSAiC field campaign. The results are highly relevant and represent a very good data set needed for better understanding of radar returns from sea ice and the snow layer. From a technical perspective I believe the methods and data are quite sound and thoroughly described in the paper. However, the paper stops short of providing information that would be of most use as a reference to understanding or improving retrieval techniques from satellite or airborne altimeters.

The authors thank the reviewer for their time and effort in reviewing our manuscript and for the constructive feedback. Please find our detailed point by point responses to the comments below.

What would be most useful here would be some results and discussion showing the observed difference in backscatter between the snow-air and snow-ice interfaces as well as the variability that was seen.

The primary objective of this paper was to a) introduce the instrument, and its deployment on sea ice during the MOSAiC expedition, both in the altimeter and scatterometer mode; and b) to demonstrate examples of radar signatures from both these modes, to ultimately demonstrate the potential of this instrument to retrieve snow depth on sea ice. Detailed analysis (including linking fully-polarimetric radar scattering, to snow/sea ice geophysical property measurements) will be further investigated in the upcoming manuscript, and including (as mentioned) relative contributions of the surfaces observed to backscatter in both bands. A quick look at the power vs range can be seen in Figure 8 where the relative power values are shown plotted against range (depth) at different locations along the transect and for both the northern and southern transects. This demonstrates some of the variability seen between transects.

Tying these to some of the physical properties measured in-situ such as salinity would also be useful. It is mentioned that this paper represents a first data set and that further studies will follow, from that perspective I do think the paper is indeed quite useful and should be published. But it would be nice to have the results placed in a bit better context towards how they could specifically be used in understanding satellite or airborne radar returns.

We agree with the reviewer comment to add ice salinity information to the analysis. In the revised version of the manuscript, we have added snow/sea ice salinity results for preliminary analysis (see lines 320-325; 450-452), as follows:

"During leg 1, sea ice thickness measurements made via drill holes ranged between 80 and 96 cm. At the start of leg 2, ice thickness at the third established RS site was 92cm, increasing to 135 cm (29 January). Measurements of sea ice freeboards during leg 2 ranged between 7 and 10cm. Ice cores revealed overall low salinity (< 1ppt), until the few centimetres above to the ice/water interface, where salinities increased between 6 and 8 ppt. The upper 20cm of the ice was relatively consistent in its low salinity (0 – 0.5 ppt), which was comprised of refrozen melt ponds."

"Snow and SYI properties from the northern transect were found to be similar to the three RS sites. Snow at the RS sites was consistently dry, cold (bulk snow temperature ~ -25°C from all RS sites), and brine-free."

Line 280: The MOSAiC floe description seems a bit simplistic with declarations about the floe properties which ignore some of the variability within the floe itself. I would suggest referencing the paper by Krumpen et al. on the floe description and history.

Krumpen, T., Birrien, F., Kauker, F., Rackow, T., von Albedyll, L., Angelopoulos, M., Belter, H. J., Bessonov, V., Damm, E., Dethloff, K., Haapala, J., Haas, C., Harris, C., Hendricks, S., Hoelemann, J., Hoppmann, M., Kaleschke, L., Karcher, M., Kolabutin, N., Lei, R., Lenz, J., Morgenstern, A., Nicolaus, M., Nixdorf, U., Petrovsky, T., Rabe, B., Rabenstein, L., Rex, M., Ricker, R., Rohde, J., Shimanchuk, E., Singha, S., Smolyanitsky, V., Sokolov, V., Stanton, T., Timofeeva, A., Tsamados, M., and Watkins, D.: The MOSAiC ice floe: sediment-laden survivor from the Siberian shelf, The Cryosphere, 14, 2173–2187, https://doi.org/10.5194/tc-14-2173-2020, 2020.

We have added Krumpen et al. (2020) reference describing the MOSAiC floe description and its history in the revised manuscript (see lines 289-293), as follows:

"The MOSAiC Central Observatory (CO) around the German research vessel R/V Polarstern was established on an oval shaped ice floe approximately 3.8 km by 2.8 km, located north of the Laptev Sea (85°N 136°E). The floe was formed north of New Siberian Islands, via a polynya event, in the beginning of December 2018 (Krumpen et al., 2020). This floe underwent extensive weathering and survived the 2019 summer melt, was heavily deformed, and consisted of predominantly remnant second-year ice (SYI)."

Some additional description of the ice properties beneath the radar site would also be helpful here and in other sections of the text such as Line 290 when it is brought up that the radar was moved.

We agree with the reviewer comment to add ice salinity information to the analysis. In the revised version of the manuscript, we have added snow/sea ice salinity results for preliminary analysis (see lines 317-325; 449-450), as follows:

"During leg 1, sea ice thickness measurements made via drill holes ranged between 80 and 96 cm. At the start of leg 2, ice thickness at the third established RS site was 92cm, increasing to 135 cm (29 January). Measurements of sea ice freeboards during leg 2 ranged between 7 and 10cm. Ice cores revealed overall low salinity (< 1ppt), until the few centimetres above to the ice/water interface, where salinities increased between 6 and 8 ppt. The upper 20cm of the ice was relatively consistent in its low salinity (0 – 0.5 ppt), which was comprised of refrozen melt ponds."

"Snow and SYI properties from the northern transect were found to be similar to the three RS sites. Snow at the RS sites was consistently dry, cold (bulk snow temperature ~ -25°C from all RS sites), and brine-free."

Line 900-905: What does it mean that "the power that comes from above the air/snow interface within a few cm of the peak is simply the impulse response of the radar"? I'm not sure why the peak is shifted several cm, and if it is the impulse response of the radar then why does it look so different than the metal plate and calibration examples? Perhaps instead it reflects the surface height distribution and scattering characteristics of the surface.

The peak in the calibration plot has been shifted in range and power to fit the peak observed from the metal plate (we have added text to explain this.) We would expect the peak from the exposed snow surface to be seen at an increased range for two reasons: 1) The plate will always sit on top of the highest topography and has a finite thickness of ~ 2 cm, so its top surface is closer to the antennas; and 2) the plate does not cover the radar footprint. Hence, the power returned from the plate, which dominates the return, is from closer to the nadir point than the exposed snow surface. We have added text in the revised manuscript (See lines 416-420) to explain this, as follows:

"We would expect this because the metal plate, approximately 15 × 55 cm in size, did not fill the entire footprints of the Ka- and Ku-band antennas, and the plate sits atop the highest points on the snow surface and has a finite thickness of ~ 2 cm. Therefore, its surface appears closer than the snow surface as it dominates the return: the measured peak range of the metal plate of 1.53 m; when the plate is removed, the air-snow peak appears at about 1.55 m at both frequencies. The relative power is also much lower because the snow scatters light in more heterogeneous directions than the metal plate"

Figure 7 and other figures: Does the Range from Antenna represent the range assuming a speed of light in free space or in snow?

Throughout the paper, the range is shown assuming a speed of light in free space, we have added text to clarify this on line 431-432, as follows:

"Results are shown as both the radar range from antenna (in meters) along with the VV power (in dB) along a short transect distance; all radar range data in this paper are shown scaled with radiation propagating at the velocity of light in free space."

Figure 9 and Line ~440: This figure and discussion are very interesting and is highly relevant for assumptions about the radar backscatter from the snow-air and snow-ice interfaces from altimeters. However, this is very qualitative and hard to put the results in a useful context as presented here. It would be useful to show at the least the backscatter difference between the snow-air interface from the algorithm and the value from the magnaprobe location.

We are pleased at the reviewer's comment about the relevance to altimetric data and are also very interested in exploring the potential for these data to provide perspectives for satellite radar altimetry – the results so far have been highly motivating and show exciting potential. However, as mentioned earlier, the primary objective of this paper was to a) introduce the instrument, and its deployment on sea ice during the MOSAiC expedition, both in the altimeter and scatterometer mode; and b) to demonstrate examples of radar signatures from both these modes, to ultimately demonstrate the potential of this instrument to retrieve snow depth on sea ice. Detailed analysis (including linking fully-polarimetric radar scattering, to snow/sea ice geophysical property measurements) will be further investigated in the upcoming manuscript. We have included plots to demonstrate this potential, but a separate publication will be required to explore the analysis in detail including combination of datasets, data processing and detailed results which are beyond the scope of this paper. Nevertheless, we have now added the numerical value of the backscatter from the snow-air interface as detected from the algorithm and also the value of the backscatter from where the magnaprobe hit the snow/ice interface. This is on Line 493-495 where we state:

"Overall, the mean power at the air/snow interface (as picked by the algorithm) is -31 and -20 dB for the Ka- and Ku-band, respectively, both with a standard deviation of 3 dB. The mean power at the MagnaProbe-derived snow depths is -45 and -30 dB for the Ka- and Ku-band, respectively, with standard deviation of 6 dB."

---

## Author Comment (AC2) · 14 Oct 2020

**Response to the Reviewer #2**

This work describes a novel ground-based dual-frequency radar used to evaluate interactions with snow on sea ice at frequencies common to CryoSat-2, AltiKa, and their follow-on mission CRISTAL. Microwave interactions with snow are discussed as a key uncertainty limiting accuracy of altimetry-based freeboard and a potential source of snow depth information leveraging frequency dependent interaction. Stationary scans and transect measurements completed as part of MOASiC highlight complex interactions and a multitude of causal mechanisms. An extensive set of snow measurements are introduced, which in the future, will be used to decompose variations in penetration and scattering. The work focuses on introducing dual-frequency theory, the KuKa radar system, and provides case-studies to evaluate spatial, temporal, and angular dependency of the observed environment.
Overall, the work contributes highly relevant results to the remote sensing of sea ice and the datasets described will to be critical to ongoing retrieval development. The case study examples provide a good variety of configuration and observed conditions, illustrating many of the complexities involved. However, I felt the system descriptions and quantitative backscatter analysis should be strengthened prior to publication. For the system description, the reader is directed to previous literature to establish methods for calibration and processing. Given advancements of the current system (larger bandwidth, transect profiling, multiple frequencies), additional information is needed to understand system uncertainty with respect to the relative backscatter and NRCS analysis:

The authors thank the reviewer for their time and effort in reviewing our manuscript and for the constructive feedback. Please find our detailed point by point responses to the comments below.

Calibration procedures for NRCS and relative backscatter: On lines 276 and 373 corner reflector calibration is mentioned and methods are attributed to those introduced in Geldsetzer et al. (2007) and King et al. (2013). The referenced studies indicate that external targets and background noise estimates were necessary for absolute calibration of NRCS. A factor for this is applied in Eqn (3) of the supplement but it is unclear what is used to accomplish the calibration. In depth analysis of system uncertainty is beyond scope, but an indication of system stability throughout the campaign or between scan types is needed if magnitudes are to be compared across extended temporal periods or between sites.

An external calibration was separately carried out for NRCS and polarimetric quantities, conducted at the RS site on 16th January 2020, using a trihedral corner reflector. The supplementary material describes the steps followed to calculate the calibration process. Regarding the long term system stability, the internal calibration loop tracks any gain variations except for those components which are outside the calibration loop, including the cables to the antenna and the antenna ports on the switches. This is why, we perform occasional corner reflector calibration when the instrument is deployed in different environments. The instrument manufacturer recommends external calibration once per deployment, to avoid instrument drifting due to hardware failure. We have included detailed information on the calibration process and long-term system stability in the revised manuscript (See lines 263-269), as follows:

"No near-field correction is applied, since the antenna far-field distance is about 1 m. An external calibration was separately carried out for calculating radar cross section per unit area (NRCS) and polarimetric quantities, conducted at the RS site on 16th January 2020, using a trihedral corner reflector positioned in the antenna's far-field (~ 10 m). In regards to long-term stability, the internal calibration loop tracks any gain variations, including the cables to the antenna and the antenna ports on the switches. Periodic calibration checks were performed with the corner reflector. Detailed description of polarimetric calibration procedure is provided in the Supplemental Material, following Sarabandi et al. (1990), and adopted in Geldsetzer et al. (2007) and King et al. (2013)."

Estimation of NRCS: Figure 11 introduces estimates of backscatter as a function of incidence angle, but I was unable to determine how the integrated impulse range was defined. I would expect a dynamic approach where the ground projected footprint is elongated with range and there is frequency / spatial dependency of penetration. How was impulse range addressed in determining NRCS and was it identical for both frequencies?

To compute Ku- and Ka-band NRCS, we assume that all scattering is from the surface. We compute the illuminated scene by assuming an ellipse on the surface defined by the Ku- and Ka-band antenna beamwidth. However, since the range resolution is very fine, we sum the return power over many range gates in the region of the peak, usually starting with the first range gate at a level ~10-20 dB below the peak at nadir or near-range and ending at a similar level on the far-range side of the peak. The dominant contribution to the total power are those points within ~ 10 dB of the peak, therefore, the exact threshold level for beginning and ending the integration is not critical. This process should give the same power as would have been measured with a coarse range resolution system having a single range gate covering the entire illuminated scene. From the averaged power profiles, the Ku- and Ka-band NRCS is calculated following (Sarbandi et al., 1990), and given by the standard beam-limited radar range equation

$$\text{NRCS } \sigma^0 = \frac{8\ln(2)h^2\sigma_c}{\pi R_C^4 \theta_{3dB}^2 \cos(\theta)} \left(\frac{\tilde{P}_r}{\tilde{P}_{rc}}\right)$$

Where $h$ is the antenna height, $R_C$ is the range to the corner reflector, $\theta_{3dB}$ is the antenna's one-way half-power beamwidth, and $\tilde{P}_r$ and $\tilde{P}_{rc}$ are the recorded power from the illuminating scene and the corner reflector, respectively. The process is same for both frequencies, although the antenna footprints are not identical. This information is now added in the revised manuscript (See lines 373-384).

Discussion of errors: Errors for NRCS are described briefly on line 485 as being plus minus 1.7 dB or greater. What sorts of challenges might this present in analysis and are these errors stable between configurations and deployments? I am also interested in understanding if the errors described for NRCS (Figure 11) are valid for the individual or mean waveform analysis given range dependent noise demonstrated in Figure 6.

The error estimates only provide a conservative estimate for the total NRCS error. The primary sources of error in NRCS estimate arise from calibration error (multiplicative bias; due to presence of metal tripod supporting the trihedral reflector), usage of finite SNR, standard deviation in estimated signal power (random; as a function of number of independent samples and noise samples, and finite SNR), and errors due to approximations used for target range, incidence angle

and scan footprint area. We have added text in the revised manuscript (See lines 535-543) to explain this, as follows:

"The KuKa radar demonstrates and maintains a high SNR across a large range of θ, gradually decreasing with increasing θ. At nadir, the co-polarized SNRs are observed to be ~ 85 dB (Ka-band) and ~ 65 dB (Ku-band), while at far-range θ, SNRs decrease to ~ 80 dB (Ka-band) and ~ 55 dB (Ku-band). These ranges are consistent for measurements acquired during the cold and warm periods on 10 and 15 November, respectively. Even though, system error can influence the observed Ku- and Ka-band backscatter variability, spatial variability of the snow surface within the radar footprint may also add to the error estimates, especially at steep θ with lower number of independent samples."

With respect to backscatter analysis, the presented interpretations could be enhanced with additional quantitative analysis. For example, Figure 9 and the associated text describes a good agreement with the snow depth, but it is unclear how this agreement was determined. Extraction of relative backscatter at the identified interfaces might be useful when associating peaks identified in the mean profiles (Figure 8).

We agree with the reviewer that this merits additional quantitative analysis, and this is currently underway. This analysis will be published in a separate paper which will detail the methodologies and results. The primary objective of this paper was to a) introduce the instrument, and its deployment on sea ice during the MOSAiC expedition, both in the altimeter and scatterometer mode; and b) to demonstrate examples of radar signatures from both these modes, to ultimately demonstrate the potential of this instrument to retrieve snow depth on sea ice. This plot demonstrates this potential. We agree that this paper also needs a description of the processing done to produce Figure 9 and we have added this in the text and figure caption to explain how this was done. Specially, lines 482 to 497 in the revised manuscript now state:

"These VV (and HH) data demonstrate the potential for detailed comparisons between KuKa data and coincident datasets such as snow MagnaProbe and SMP to explore the scattering characteristics in the Ka- and Ku-bands, over varying snow and ice conditions. Further insight is gained by overlaying the MagnaProbe snow depth (**Figure 9** for the northern transect). To make this comparison, both the KuKa and MagnaProbe data have been corrected using the FloeNavi script developed by Hendicks (2020), which converts latitude, longitude and time data into floe coordinates, referenced to the location and heading of the Polarstern ship. The data along the transect were then divided into 5 m sections, and in each section the snow depth (from the MagnaProbe), Ku-band echoes and Ka-band echoes were averaged and plotted as shown in Figure 9 which shows the averaged echoes with average snow depths overlaid. Also shown is the first peak identified using a simple peak detection method that corresponds to the snow/air interface. Of note is that there appears to be agreement between the first peaks detected in the Ka- and Ku-bands, and between peaks in the Ku-band echoes and the MagnaProbe snow depths (which have been scaled by 0.8 to take into considering the slower wave propagation speed into the snow). Overall, the mean power at the air/snow interface (as picked by the algorithm) is -31 and -20 dB for the Ka- and Ku-band, respectively, both with a standard deviation of 3 dB. The

mean power at the MagnaProbe-derived snow depths is -45 and -30 dB for the Ka- and Ku-band, respectively, with standard deviation of 6 dB. The mechanisms whereby the $\sigma_{VV}^0$ increases at the snow/ice interface, and correlations between snow depth and these peaks, will be further investigated and quantified in a publication which will analyse these data in detail."

Finally, the acknowledgements section indicates that data are available at the UK Polar Data Centre but I was unable to retrieve them using common terms from the paper. If the data are available a set of links would be helpful.

We are awaiting the final doi from the UK Polar Data Center, and this should hopefully be remedied before final publication.

Thanks to the authors for their dedicated work to develop and deploy a novel radar system in this challenging environment. The datasets and analysis are valuable contributions to community and will improve understanding snow-radar interactions on sea ice. Please find my specific comments indicated below with page and line numbers.

The authors thank the reviewer for the compliment and constructive feedback. Please find our detailed point by point responses to the comments below.

Specific comments P5 L139 – The seasonal aspect is an important distinction from previous ground-based Ku-band studies on sea ice. Could a table 3 be amended or a description be added to inform the reader of how the preliminary dataset fits in the context of the larger seasonal acquisitions?

At the moment we only have access to the autumn/winter observations, and we will be

evaluating the summer/autumn data once the ship returns to Bremerhaven and the data are

uploaded to the MOSAiC central data storage. We now state at the end of the introduction (See

lines 147-149), as follows: "This preliminary study fits well within the context of conducting a

larger seasonal analysis of coincident Ka- and Ku-band radar signatures and its evolution over

snow-covered sea ice from autumn freeze-up through winter, to melt-onset thermodynamic

regime, once all data collected during the MOSAiC campaign become available."

P7 L218 – Questions regarding calibration: Has absolute (external) calibration of the system been completed or is the calibration process solely internal? The supplementary materials state an absolute calibration was completed and P9 L275 mentions a corner reflector, but I had difficulty determining the process or how often it is required. Please include information on how external factors (changes in cabling or mounting structure) were accounted for in the calibration process.

An external calibration was separately carried out for NRCS and polarimetric quantities, conducted at the RS site on 16[th] January 2020, using a trihedral corner reflector positioned in the antenna's far-field (~ 10 m). The supplementary material describes the steps followed to calculate the calibration process. Regarding the long-term system stability, the internal calibration loop tracks any gain variations except for those components which are outside the calibration loop, including

the cables to the antenna and the antenna ports on the switches. This is why, we perform occasional corner reflector calibration when the instrument is deployed in different environments. The instrument manufacturer recommends external calibration once per deployment, to avoid instrument drifting due to hardware failure. We have included detailed information on the calibration process and long-term system stability in the revised manuscript (See lines 263-269), as follows:

"No near-field correction is applied, since the antenna far-field distance is about 1 m. An external calibration was separately carried out for calculating radar cross section per unit area (NRCS) and polarimetric quantities, conducted at the RS site on 16th January 2020, using a trihedral corner reflector positioned in the antenna's far-field (~ 10 m). In regards to long-term stability, the internal calibration loop tracks any gain variations, including the cables to the antenna and the antenna ports on the switches. Periodic calibration checks were performed with the corner reflector. Detailed description of polarimetric calibration procedure is provided in the Supplemental Material, following Sarabandi et al. (1990), and adopted in Geldsetzer et al. (2007) and King et al. (2013)."

P7 L225 – Can you add a description of the physical separation between the two horns? This is needed to reproduce the overlap estimates.

Description of physical separation between the Ku- and Ka-band antenna horns are included in the revised manuscript (See lines 227-228), as follows:

"The antennas of each radar are dual-polarized scalar horns with a beamwidth of 16.5° at Ku-band and 11.9° at Ka-band, with a center-to-center spacing of 13.36 cm (Ku-band) and 7.65 cm (Ka-band)."

P8 L244 – Given that the system is operated on drifting ice have steps have been taken to compensate for ice motion when applying velocity or displacement thresholds? Please include a statement on uncertainty associated with ice motion and properties of the GPS.

Ice motion has not been compensated for in applying velocity thresholds because the ice motion (we have noted this reach a magnitude of 8 m/minute or 0.13m/s) is much less than the threshold of 0.4 m/s. The exception is in our analysis to produce Figure 9, where it was necessary in order to compare the KuKa and MagnaProbe datasets, and an explanation of this has been added the text on Figure 9 (see Lines 485-488).

"To make this comparison, both the KuKa and MagnaProbe data have been corrected using the FloeNavi script developed by Hendricks (2020), which converts latitude, longitude and time data into floe coordinates, referenced to the location and heading of the Polarstern ship."

P9 L258 – On P12 L374 it is stated that a near-field correction is applied but this line states it was not necessary. Please clarify.

No near-field correction is applied, since the antenna far-field distance is about 1 m (using $2D^2/\lambda$ formula, where D is the antenna diameter and $\lambda$ the wavelength). We have removed the near-field correction phrase) in the revised manuscript.

P9 L272 – Is the minimum threshold of 1/2 the antenna diameter greater than the accuracy of the GPS?

No, but the precision of the GPS shows changes on this size scale, so we can tell if the sled has moved enough to generate a new independent sample.

P10 L295 – A statement on standard snow pit protocols would be a helpful addition here. I read this currently as the SMP being a primary tool. It should be noted that the SMP provides a profile of micromechanical properties and requires a statistical model to derive microstructure. Presumably, the snow castings for uCT are to be used as a high-quality reference but this is not clear.

Detailed snow/sea ice geophysical property observations were obtained as close as possible to the RS site, via weekly snow pits, bi-weekly snow depth measurements (around each RS instrument) and collection of occasional ice cores. These observations included snow specific surface area (SSA), the scatter correlation length (Proksch et al., 2015) and density made using a SnowMicroPen (SMP), snow/air and snow/ice interface temperatures with a temperature probe, snow salinity with a salinometer and SWE using a 50 cm metal ETH tube together with a spring scale. In case of hard crusts too hard for the SMP to work, snow density was collected using a density cutter. The information is added to the revised manuscript (See Lines 307-313).

"Detailed snow/sea ice geophysical property observations were obtained as close as possible to the RS site, via weekly snow pits, bi-weekly snow depth measurements (around each RS instrument) and collection of occasional ice cores. These observations included snow specific surface area (SSA), the scatter correlation length (Proksch et al., 2015) and density made using a SnowMicroPen (SMP), snow/air and snow/ice interface temperatures with a temperature probe, snow salinity with a salinometer and SWE using a 50 cm metal ETH tube together with a spring scale. In case of hard crusts too hard for the SMP to work, snow density was collected using a density cutter."

P10 L 311: Sturm and Holmgren, 2017 can be used as a citation for the instead of the patent: Sturm, M., & Holmgren, J. (2017). An Automatic Snow Depth Probe for Field Validation Campaigns. Water Resources Research, 9695–9701. DOI: 10.1029/2018WR023559@10.1002/(ISSN)1944-7973.SNOWEX1

Citation replaced in the revised manuscript.

Figure 2 and P10 L312: I was not able to determine in Figure 2 where the North and South transects were completed. Could these be added?

Transect route added to Figure 2.

Figure 5 and P11 Line 343: The snow depth distributions are shown as a frequency count. Are they exact repeats in terms of the number of observations and locations?

Snow depth distributions are from as close to the same locations as possible when walking the transect loops in the dark. Total sample sizes differed slightly between dates as expected. Sample sizes for the northern transect ranged from 1014-1126 and for the southern transect from 1146-1405.

Does the internal calibration loop agreement with the metal plate suggest the system is well constrained and does not need external calibration? A clear statement on system stability would provide confidence that the measurements can be compared between deployments and configurations. Alternatively, a statement of why external calibration is not needed could be provided.

No, the agreement between the internal calibration loop and metal plate simply shows that the radar impulse response is the same whether the signal passes through the antenna and associated cables or just through the internal calibration loop. So, from this, we can state that the antenna and cables are not distorting the impulse response, which could happen if the antenna gain was drastically different between the low and high ends of the frequency sweep.

An external calibration was separately carried out for NRCS and polarimetric quantities, conducted at the RS site on 16$^{th}$ January 2020, using a trihedral corner reflector positioned in the antenna's far-field (~ 10 m). Regarding the long-term system stability, the internal calibration loop tracks any gain variations except for those components which are outside the calibration loop, including the cables to the antenna and the antenna ports on the switches. This is why we perform occasional corner reflector calibration when the instrument is deployed in different environments. The instrument manufacturer recommends external calibration once per deployment, to avoid instrument drifting due to hardware failure. We have included detailed information on the calibration process and long-term system stability in the revised manuscript (See Lines 263-269).

P13 L390 and P13 408 411: Can quantities be associated with the terms dominate and significant? For example, what % of the signal is coming from the AS interface at each frequency or what % of backscatter is observed at ranges beyond where the SI interface is presumed to be.

These quantities are not constant along the transect and are highly dependent on the snow and ice sampled. For our follow on paper with the detailed analysis of these data we are looking at this for the full KuKa bandwidth as well as resampling at CryoSat-2 and Altika bandwidths to understand these relative contributions, however, this requires further work and is outside the scope of this paper which demonstrates the capability of the instrument to gather data to enable this. We show (Figure 8) averaged power-range plots for shorter sections of the transects, where the relative power returned from each surface can be seen.

P14 L425: How are location of the peaks determined? Its hard to pick where the peaks are by eye beyond the main lobe in Figure 8.

The peak locations noted here are the locations of maxima as shown in the data in Figure 8. Further analysis is needed to determine correspondences between the location of these maxima and the physical snow and ice volumes/boundaries hence we tentatively suggest possible sources of the apparent scattering surfaces but do not yet make firm conclusions. The panels which contain the Ka- and Ku-band data are explained more clearly through an update to the caption of Figure 8.

P14 L434: Does the cross-pol ratio support the presence of increased multiple scattering for the Northern Transect?

We believe the reviewer is asking whether the HV data show higher power values for the Northern transect than the Southern transect. The HV data are beyond the scope of this paper and show variability along the transects and hence will require detailed analysis to determine whether this is the case.

P14 L441: Peak detection = greatest magnitude? This is a critical point of clarification as several studies have suggested that surface bedforms and/or roughness dictate where along the rising edge the AS interface can be found.

We agree with the reviewer that the power returned will depend on the surface characteristics – as for the snow/ice interface. Detection of the air/snow interface requires a range of approaches to understand how to best characterise the appearance of this interface in the KuKa data, as noted here we are using a simple algorithm to detect the peak for this paper introducing the instrument. We are currently undergoing more detailed analysis and trying approaches including edge detection to determine how best to detect and characterise the echoes and will report this in our following publication with further detailed analysis.

P14 P442 and Figure 9: What is used to define agreement between the Ku-band signal and snow depth? Please provide quantities to support.

For this paper, we are demonstrating the potential of the instrument to detect snow depth, and hence have included a demonstration of the visual agreement between these. We are currently analysing the data in more detail, determining the best methods to investigate and quantify the agreement and will include detailed discussion of this in our future publication. However, we now add on Lines 483-498:

"These VV (and HH) data demonstrate the potential for detailed comparisons between KuKa data and coincident datasets such as snow MagnaProbe and SMP to explore the scattering characteristics in the Ka- and Ku-bands, over varying snow and ice conditions. Further insight is gained by overlaying the MagnaProbe snow depth (**Figure 9** for the northern transect). To make this comparison, both the KuKa and MagnaProbe data have been corrected using the FloeNavi script developed by Hendricks (2020), which converts latitude, longitude and time data into floe coordinates, referenced to the location and heading of the Polarstern ship. The data along the transect were then divided into 5 m sections, and in each section the snow depth (from the MagnaProbe), Ku-band echoes and Ka-band echoes were averaged and plotted as shown in Figure 9 which shows the averaged echoes with average snow depths overlaid. Also shown is the first peak identified using a simple peak detection method that corresponds to the snow/air interface. Of note is that there appears to be agreement between the first peaks detected in

the Ka- and Ku-bands, and between peaks in the Ku-band echoes and the MagnaProbe snow depths (which have been scaled by 0.8 to take into considering the slower wave propagation speed into the snow). Overall, the mean power at the air/snow interface (as picked by the algorithm) is -31 and -20 dB for the Ka- and Ku-band, respectively, both with a standard deviation of 3 dB. The mean power at the MagnaProbe-derived snow depths is -45 and -30 dB for the Ka- and Ku-band, respectively, with standard deviation of 6 dB. The mechanisms whereby the $\sigma_{VV}^0$ increases at the snow/ice interface, and correlations between snow depth and these peaks, will be further investigated and quantified in a publication which will analyse these data in detail."

P15 L463: There needs to be some explanation about how estimates of NRCS were generated. What impulse range is integrated? Is the same for both systems? Does it change with incidence angle?

To compute Ku- and Ka-band NRCS, we assume that all scattering is from the surface. We compute the illuminated scene by assuming an ellipse on the surface defined by the Ku- and Ka-band antenna beamwidth. However, since the range resolution is very fine, we sum the return power over many range gates in the region of the peak, usually starting with the first range gate at a level ~10-20 dB below the peak at nadir or near-range and ending at a similar level on the far-range side of the peak. The dominant contribution to the total power are those points within ~ 10 dB of the peak, therefore, the exact threshold level for beginning and ending the integration is not critical. This process should give the same power as would have been measured with a coarse range resolution system having a single range gate covering the entire illuminated scene. From the averaged power profiles, the Ku- and Ka-band NRCS is calculated following (Sarbandi et al., 1990), and given by the standard beam-limited radar range equation

$$\text{NRCS } \sigma^0 = \frac{8\ln(2)h^2\sigma_c}{\pi R_C^4 \theta_{3dB}^2 \cos(\theta)}\left(\frac{\tilde{P}_r}{\tilde{P}_{rc}}\right)$$

Where $h$ is the antenna height, $R_C$ is the range to the corner reflector, $\theta_{3dB}$ is the antenna's one-way half-power beamwidth, and $\tilde{P}_r$ and $\tilde{P}_{rc}$ are the recorded power from the illuminating scene and the corner reflector, respectively. The process is same for both frequencies, although the antenna footprints are not identical.

This information is now added in the revised manuscript (See lines 373-384) as discussed above.

P16 L510: There appears to be little or no difference at angles common to altimetry (near nadir figure 11e) between the two dates. How should this finding be interpreted for current and upcoming missions?

Although the co-pol phase difference does not indicate any changes at nadir or lower end of the near-range incidence angles, there is a demonstrable steep increase in both Ka- and Ku-band backscatter between the cold and the warm day. This has already been described in the original manuscript (refer P15 L467-474). In regards to implications for current and upcoming altimeter and SAR missions, we have already described them in the original manuscript (P17 L523-535). With regards to the low sensitivity of the co-pol phase difference (derived polarimetric parameter) to snow surface warming and/or snow surface topography changes, at nadir and near-range angles, our preliminary observation is that, larger anisotropic scattering from the snow pack are more

prominent at shallow incidence angles, and results in second- or multiple-order scattering mechanisms within the snow pack. However, further research is required to confirm this behavior and is beyond the scope of this paper.

---

## Author Response (AR2)

We thank the editor for their final editing of our manuscript and have made all changes as recommended (tracked changes here)

[revised manuscript text omitted]